# Towards Interpretable Controllability in Object-Centric Learning

## Abstract

The binding problem in artificial neural networks is actively explored with the goal of achieving human-level recognition skills through the comprehension of the world in terms of symbol-like entities. Especially in the field of computer vision, object-centric learning (OCL) is extensively researched to better understand complex scenes by acquiring object representations or *slots*. While recent studies in OCL have made strides with complex images or videos, the interpretability and interactivity over object representation remain largely uncharted, still holding promise in the field of OCL. In this paper, we introduce a novel method, Slot Attention with Image Augmentation (SlotAug), to explore the possibility of learning interpretable controllability over slots in a self-supervised manner by utilizing an image augmentation strategy. We also devise the concept of sustainability in controllable slots by introducing iterative and reversible controls over slots with two proposed submethods: Auxiliary Identity Manipulation and Slot Consistency Loss. Extensive empirical studies and theoretical validation confirm the effectiveness of our approach, offering a novel capability for interpretable and sustainable control of object representations. Code will be available upon acceptance.

## 1 Introduction

Compositional comprehension of visual scenes (Marr, 2010; Tenenbaum et al., 2011; Johnson et al., 2017; Fischler & Elschlager, 1973), essential for various computer vision tasks such as localization (Cho et al., 2015) and reasoning (Mao et al., 2019), requires human-like understanding of complex world (Treisman, 1996; Spelke & Kinzler, 2007; Lake et al., 2017). In response to this, *object-centric learning* (OCL) has emerged as an active research area (Locatello et al., 2020; Kipf et al., 2021; Greff et al., 2016). OCL aims to enable a model to decompose an image into its components in terms of an object, and to acquire object representations or *slots*, without relying on human-annotated labels.

In pursuit of a deeper understanding of images, interpretable and controllable object representation has been studied (Greff et al., 2019; Burgess et al., 2019; Singh et al., 2023). Nevertheless, the previous approaches face limitations in achieving *interpretable controllability* as they require additional processes to figure out how to interact with slots, such as exploring a connection between specific values in slots and object properties by a manual exhaustive search, and training a feature selector with ground-truth object properties (Fig. 1(a)). This issue arises due to a training-inference discrepancy, wherein interactions with slots are only considered during the inference stage. This discrepancy problem brings ambiguity in how to interact with object representations, hindering interpretable controllability. Furthermore, learning interpretable controllability is followed by another challenge: object representation should be intact even after multiple manipulations by humans. In this context, we devise the concept of *sustainability* pertaining to the ability to preserve the nature of slots, allowing for iterative manipulations; we refer to Fig. 4 and 5 for establishing the earlier motivation.

In this work, we advance the field of OCL with respect to the interpretability of object representation. To achieve interpretable controllability, we propose a method that enables the manipulation of object representation through semantically interpretable instructions in a self-supervised manner. We address the training-inference discrepancy problem by incorporating image augmentation into our training pipeline (Fig. 1(c)). By involving the slot manipulation in the training, we can resolve the discrepancy problem and streamline the way to interact with slots in the inference stage (Fig. 1(b) and (d)).

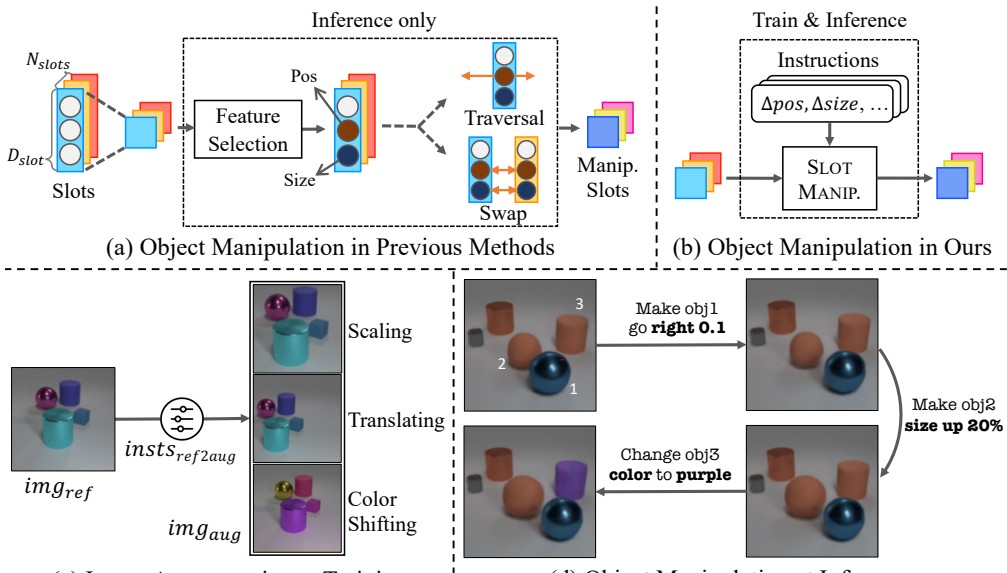

Figure 1: **Overview of our method compared to the previous methods. (a)** Previous methods require an additional process to manipulate slots such as feature selection during inference. **(b)** Our model, however, has the shared process of manipulating slots between the training and inference stages. **(c)** To ensure homogeneity between the training and inference stages, we incorporate scenarios involving image manipulation into the training phase. This includes the application of simple image augmentation techniques such as scaling, translating, and color shifting. **(d)** Upon completion of the training, our model achieves interpretable controllability, enabling users to manipulate individual objects according to their intentions.

Second, to attain sustainability in object representation, we introduce *Auxiliary Identity Manipulation* (AIM) and *Slot Consistency Loss* (SCLoss). AIM is a methodology designed to facilitate the learning of the concept of multi-round manipulation. AIM is implemented by incorporating an auxiliary manipulation process into the intermediate stage of slot manipulation, where the auxiliary manipulation introduces no semantic changes to object properties such as zero-pixel translations. This simple auxiliary process can expose our model to multi-round manipulation: we can make two-round manipulations with one instruction from the augmentation and the other from the auxiliary manipulation. Additionally, SCLoss allows our model to learn the concept of reversible manipulation, such as the relationship between moving an object 1 pixel to the right and moving it 1 pixel to the left. After being trained with SCLoss, our model produces consistent and reusable representations that can undergo multiple modifications and enhance their usability. With AIM and SCLoss, our model achieves sustainability in object representation.

Extensive experiments are shown to demonstrate the interpretable and sustainable controllability of our model. To assess interpretability, we conduct object manipulation experiments where slots are guided by semantically interpretable instructions. In evaluating sustainability, we introduce novel experiments, including the durability test. Our evaluation encompasses not only pixel space assessments such as image editing via object manipulation, but also slot space analyses such as property prediction, to provide a comprehensive examination of our approach.

**The main contributions of the paper** can be summarized as follows: (i) Towards interpretable controllability in object-centric learning, we incorporate image augmentation into the training process to explore the possibility of interpretable controllability over object representation. (ii) We introduce a novel concept of sustainability in object representation, strengthening the interaction between neural networks and humans, which is an important aspect of interpretable controllability. (iii) Two novel methods are introduced to ensure sustainability in object representation: Auxiliary Identity Manipulation and Slot Consistency Loss. (iv) We extensively validate the effectiveness of our method through empirical studies including novel experiments and analyses on pixel and slot space.

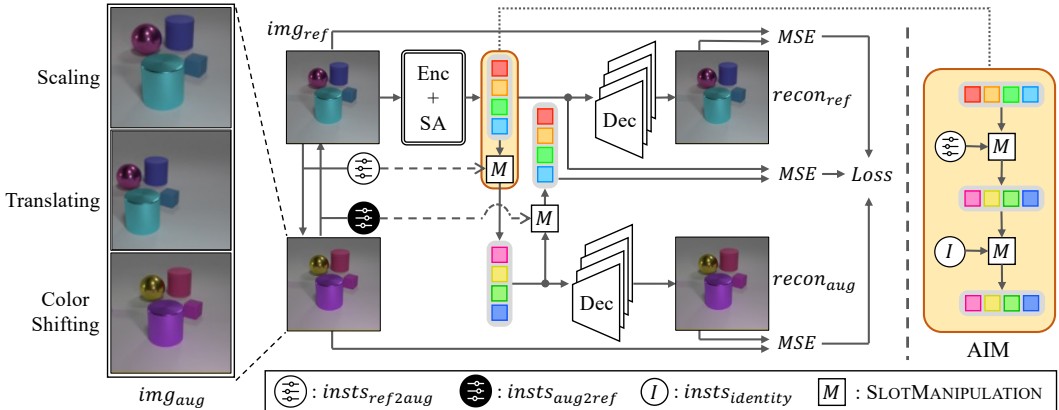

Figure 2: **Architecture of our model.** From a given image $img_{ref}$, we first generate an augmented image $img_{aug}$ (leftmost part of the figure), and the corresponding instruction $insts_{ref2aug}$ and its inverse $insts_{aug2ref}$. Our model produces slots from $img_{ref}$ and decodes the slots to reconstruct the original image ($recon_{ref}$). The slots are also manipulated with SLOTMANIPULATION module which takes $insts_{ref2aug}$ as the other input. We incorporate Auxiliary Identity Manipulation (AIM) into this manipulation process. The details are provided in the right part of the figure. The manipulated slots are then simultaneously 1) decoded into a reconstruction of the augmented image $recon_{aug}$, and 2) re-manipulated by SLOTMANIPULATION with $insts_{aug2ref}$. Our total loss consists of the reconstruction losses of reference and augmented images, and the slot-level cycle consistency.

## 2 METHODS

### 2.1 PRELIMINARY: SLOT ATTENTION

Slot Attention (SA) (Locatello et al., 2020) introduces the concept of *slots*, a set of $K$ vectors of dimension $D_{slot}$, that serves as the object representation. The slots are initialized by a Gaussian distribution with learnable mean $\mu$ and sigma $\sigma$, and are updated over $T$ iterations by the slot attention module. The final slots are then decoded to reconstruct the target image. To provide a comprehensive understanding of our method, we describe the mechanism of spatial binding (Greff et al., 2020; Treisman, 1996; Buehner & Humphreys, 2010) by Slot Attention in Alg. A, referred to as SPATIALBINDING, in the Appendix due to the space limitation. Each updated slot is then decoded individually into an RGBA image using a spatial broadcast decoder (Watters et al., 2019) which is shared across slots. The decoded images are blended into a single image using alpha masks to reconstruct the input image. The training objective is the mean squared error (MSE) between the original input image and the reconstructed image, following a self-supervised learning approach.

### 2.2 SELF-SUPERVISED LEARNING FOR INTERPRETABLE CONTROLLABILITY

**Data augmentation.** We introduce a simple data augmentation scheme that, for a given input image or a reference image $img_{ref} \in \mathbb{R}^{H \times W \times 3}$, generates an augmented image $img_{aug} \in \mathbb{R}^{H \times W \times 3}$ and the transformation instructions between them, $insts_{ref2aug}$ and $insts_{aug2ref} \in \mathbb{R}^{K \times L}$, where $L$ indicates the total number of values to represent the object properties. $img_{aug}$ is produced by a random translation, scaling, or color shifting on $img_{ref}$. To transform $img_{ref}$ into $img_{aug}$, we employ a set of instructions known as $insts_{ref2aug}$. These instructions comprise a list of values that dictate the augmentation process, including translation values, a scaling factor, and color shift values in the HSL color space. We also have the inverse instructions, $insts_{aug2ref}$, which allow us to revert $img_{aug}$ back to $img_{ref}$. Details for the data augmentation are described in the Appendix.

**Training.** We propose a novel training process that leverages image augmentation (Fig. 2). Our training scheme enables learning interpretable controllability which allows us to interact with the model via semantically interpretable instructions. Our training process involves data augmentation, spatial binding, slot manipulation, and image reconstruction via slot decoding. For a given input image,

---

**Algorithm 1** Our slot manipulation algorithm in pseudo code. The algorithm takes `slots` and `insts` as input, where `insts` contains the information for modifications. *J* represents the number of object properties, while $P_{j,f}$ and $P_{j,l}$ indicate the first and last indices of the j-th object property values. The `PropertyEncoder` outputs a vector of the same dimension as `slots`, $K \times D_{slots}$.

---

1: **function** SLOTMANIPULATION(`slots` $\in \mathbb{R}^{K \times D_{slots}}$, `insts` $\in \mathbb{R}^{K \times L}$)
2:     **for** $j = 0 \dots J$ **do**
3:         `inst`$_j$ = `insts`$[:, P_{j,f} : P_{j,l}]$
4:         `inst_vec`$_j$ = `PropertyEncoder`$_j$(`LayerNorm`(`inst`$_j$))
5:         `slots` = `slots` + `inst_vec`$_j$
6:     **end for**
7:     `slots` = `slots` + MLP(LayerNorm(`slots`))
8:     **return** `slots`
9: **end function**

---

we initially perform data augmentation to yield $img_{ref}$, $img_{aug}$, $insts_{ref2aug}$, and $insts_{aug2ref}$. Then, the model performs SPATIALBINDING on $img_{ref}$ to produce $slots_{ref}$.

Thereafter, the model conducts SLOTMANIPULATION (Alg. 1) to modify $slots_{ref}$ based on $insts_{ref2aug}$. In the SLOTMANIPULATION, we utilize a newly introduced component called *PropertyEncoder*, which is 3-layer multi-layer perceptrons (MLPs). This PropertyEncoder is responsible for generating vector representations, `inst_vec`, which capture the essence of transformation instructions. Each PropertyEncoder$_j$ generates an `inst_vec`$_j$ that encodes the values of $insts_{ref2aug}$ for the $j$-th property. These vectors are then added to $slots_{ref}$ to reflect the effect of $insts_{ref2aug}$. This addition is followed by a residual connection, along with layer normalization and another MLP to generate $slots_{ref2aug}$.

Lastly, $slots_{ref2aug}$ is decoded by the decoder to create the $recon_{aug}$, the reconstruction image for the $img_{aug}$. The MSE between $img_{aug}$ and $recon_{aug}$ serves as a training loss, $\mathcal{L}_{aug}$. To ensure stable training, we also adopt an additional loss, $\mathcal{L}_{ref}$, the MSE between the $img_{ref}$ and $recon_{ref}$, the reconstructed reference image decoded from $slots_{ref}$. Accordingly, our training loss for image reconstruction is defined as $\mathcal{L}_{recon} = \mathcal{L}_{ref} + \mathcal{L}_{aug}$.

**Inference.** To perform object manipulation, we provide the model with the position of the target object, along with the instruction to be carried out. When given the position of the target object, we use the Hungarian algorithm (Kuhn, 1955) to find the slot for the object closest to the given position. To predict the position of an object encoded in a slot, we compute the center of mass acquired from the alpha mask by the decoder or from the attention map between the visual encodings and the slot. After figuring out the desired slot, we perform slot manipulation with the given instructions.

### 2.3 SUSTAINABILITY IN OBJECT REPRESENTATION

In this work, we introduce *sustainability* which stands for the concept that object representations should sustain their integrity even after undergoing iterative manipulations. Therefore, sustainability is a key feature that contributes to the reliable and reusable object representation.

**Auxiliary Identity Manipulation (AIM)** serves as the identity operation for slot manipulation, indicating no changes in object properties. By manipulating slots with instructions that include zero values for translation, one for scaling, and so on, AIM is supposed to make each slot preserve the original identity of the object. We incorporate AIM into the training process to make the model recognize and maintain the intrinsic characteristics of objects during iterative manipulations. AIM is applied to the slot manipulation process as follows:

$$
\begin{aligned}
slots'_{ref2aug} &= f(f(slots_{ref}, insts_{ref2aug}), insts_{identity}) \\
&= f(slots_{ref2aug}, insts_{identity}),
\end{aligned}
\tag{1}
$$

where $f$ represents the SLOTMANIPULATION function, and $insts_{identity}$ is the instruction that contains the identity elements for manipulating object properties. In the followings, $slots'_{ref2aug}$ is notated as $slots_{ref2aug}$ for simplicity if not mentioned.

**Slot Consistency Loss (SCLoss)** addresses the issue of a slot diverging significantly from its original state after iterative manipulations, even when a user intends to restore the corresponding object to its original state. To implement SCLoss, we introduce $slots_{revisited}$, which is derived by executing a series of SLOTMANIPULATION operations on $slots_{ref}$ using $insts_{ref2aug}$ and $insts_{aug2ref}$. Supposed that our goal is to ensure $slots_{ref}$ and $slots_{revisited}$ have the same representation, we set the MSE between them as SCLoss. As a result, the model learns to keep the two distinct slots representing the same object as close as possible and to be robust against multiple rounds of manipulation. The equation of SCLoss, $\mathcal{L}_{\text{cycle}}$, and the total training loss, $\mathcal{L}_{\text{total}}$, are as follows:

$$\mathcal{L}_{\text{cycle}} = \frac{1}{K}\|f(f(slots_{ref}, insts_{ref2aug}), insts_{aug2ref}) - slots_{ref}\|_2^2, \qquad (2)$$

$$\mathcal{L}_{\text{total}} = w_{recon}\mathcal{L}_{\text{recon}} + w_{cycle}\mathcal{L}_{\text{cycle}}, \qquad (3)$$

where $K$ is the number of slots, $f$ is the SLOTMANIPULATION function, and $w_{recon}$ and $w_{cycle}$ are the weights for the corresponding loss.

## 3 RELATED WORKS

The binding problem in artificial neural networks (Greff et al., 2020), inspired by cognitive science (Treisman, 1996; Feldman, 2013), is a subject of active exploration, aiming to attain human-like recognition abilities by understanding the world in terms of symbol-like entities (like objects). In computer vision, object-centric learning (OCL) focuses on comprehending visual scenes by considering objects and their relationships without labeled data (Xie et al., 2022; Engelcke et al., 2021; Wu et al., 2022). MONet (Burgess et al., 2019), IODINE (Greff et al., 2019), and GENESIS (Engelcke et al., 2019) have adopted autoencoding architectures (Baldi, 2012; Kingma & Welling, 2013; Makhzani et al., 2015) to accomplish self-supervised OCL, and Slot Attention (Locatello et al., 2020) introduced the concept of slot competition, which enables parallel updates of slots with a single visual encoding and decoding stage. Recent studies have leveraged large-scale models to learn object representations in complex images (Singh et al., 2021; Seitzer et al., 2022), multi-view images (Sajjadi et al., 2022a), and videos (Kipf et al., 2021; Singh et al., 2022). Other recent works have utilized object-related inductive biases to improve the OCL models. SLASH (Kim et al., 2023) addressed the instability in background separation using a learnable low-pass filter to solidify the object patterns in the attention maps. SysBinder (Singh et al., 2023) introduced a factor-level slot, called block, to disentangle object properties and enhance the interpretability in OCL.

Several studies have shown the possibility of interacting with object representation to manipulate the objects. VAE-based models such as IODINE (Greff et al., 2019) and Slot-VAE (Wang et al., 2023) showed that adjusting the values of slots can change object properties. SysBinder (Singh et al., 2023) demonstrated that replacing factor-level slot, called block, between slots exchanges the corresponding properties. However, these works have difficulties in determining ways to interact with slots as they require manual efforts to identify the features associated with specific properties. ISA (Biza et al., 2023) incorporates spatial symmetries of objects using slot-centric reference frames into the spatial binding process, enhancing interactivity of object representation for spatial properties such as position and scale. Meanwhile, our method itself has no constraint on the types of the target property, showing its expandability toward extrinsic properties such as the shape and material of objects if there exist proper image augmentation skills or labeled data. In another direction, MulMon (Li et al., 2020) and COLF (Smith et al., 2022) showed the manipulation of extrinsic object properties, such as position and z-axis rotation, by utilizing a novel view synthesis with a multi-view dataset. In contrast, our work accomplishes direct and interpretable controllability over object representation in single-view images without requiring multi-view datasets.

## 4 EXPERIMENTS

**Datasets.** We evaluate models on four multi-object datasets: Tetrominoes (Rishabh et al., 2019), CLEVR6 (Johnson et al., 2017), CLEVRTEX6 (Karazija et al., 2021) and PTR (Hong et al., 2021). For Tetrominoes, we use 60K and 15K samples as a train and a test set, respectively. CLEVR6 is a subset of CLEVR dataset, where 6 stands for the maximum number of objects in a scene. We use 35K samples for training and 7.5K samples for testing. CLEVRTEX6 is a subset of CLEVRTEX which

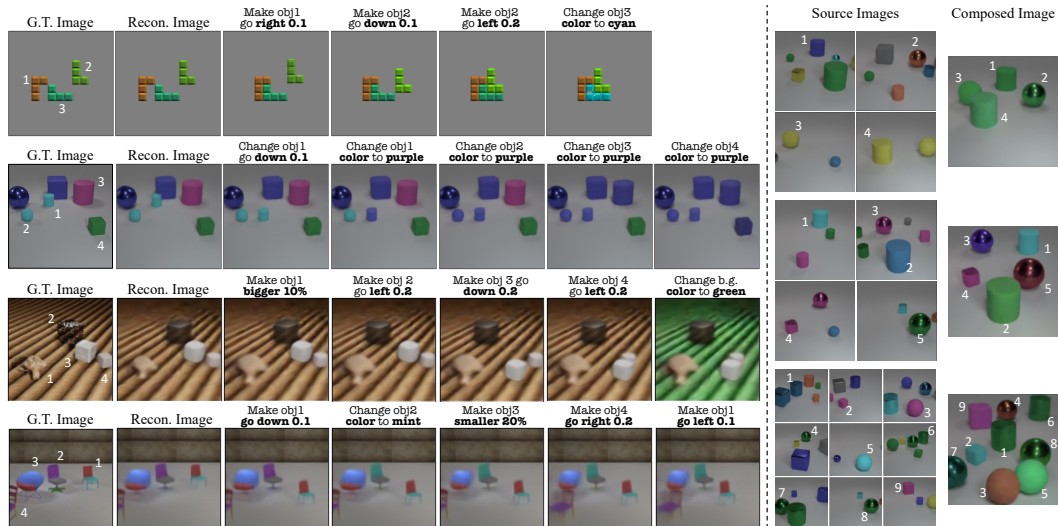

(a) Object Manipulation with Human-Interpretable Instruction      (b) Cond. Image Composition

Figure 3: **(a) Object manipulation with human-interpretable instruction.** The first and second columns are the ground-truth and reconstruction images, respectively. The following columns are the results of the controls along the instructions. Here, instructions are described with the text for easy understanding. The actual instantiation of the instructions can be found in the Appendix. From the first row onwards, the results are for Tetrominoes, CLEVR, CLEVR, and PTR, respectively. **(b) Conditional image composition.** From given source images, we can collect specific objects, which are indicated by white numbers, and manipulate them to generate a novel image.

is a complex variant of CLEVR, having complicated shapes, textures, materials, and backgrounds. CLEVRTEX6 contains 20K and 5K samples for training and testing. PTR is a dataset consisting of 52K training and 9K test samples, containing complex objects with part-whole hierarchies.

**Training.** Unless stated otherwise, the training setup follows the methodology in Slot Attention (Locatello et al., 2020). The number of epochs is 1000 with 20 warm-ups and 200 decaying epochs. We adopt AdamW (Loshchilov & Hutter, 2019) as the optimizer. The number of slots ($K$) is set to 7 and the input image size ($H \times W$) is set to $128 \times 128$, except for Tetrominoes where $K = 7$ and $H = W = 64$. The weights for the training loss are set as $w_{recon} = 1.0$ and $w_{cycle} = 0.1$. The details of the training process including the data augmentation setting are stated in the Appendix.

**Models.** We employ the same model architecture as Slot Attention. However, for CLEVRTEX6 and PTR, which are more complex datasets, we replace the encoder with ViT (Dosovitskiy et al., 2020) pretrained by MAE (He et al., 2022) and the decoder with that of SRT (Sajjadi et al., 2022b) while using an increased size of the slot attention module. The additional details for adopting large models are described in the Appendix. To clarify the methods used in ablative studies, we categorize our model into three versions: v1, which is exclusively trained with image augmentation; v2, which improves upon v1 with AIM; v3, which extends v1 with both AIM and SCLoss. For qualitative studies, we use the v3 model.

## 4.1 HUMAN-INTERPRETABLE CONTROL OVER OBJECT REPRESENTATION

### 4.1.1 IMAGE EDITING BY OBJECT MANIPULATION

As shown in Fig. 3(a), our model can manipulate individual objects. We can control not only multiple objects in a scene but also multiple properties of an object with the specific intent of users through the instructions. In the second row of the figure, we observe that our model can restore the cropped part of the object (the pink one) when we pull the object inward after changing the color of the object. Based on this observation, we can ascertain that our object representations retain the intrinsic properties of objects seamlessly even after manipulation. Furthermore, the interpretable

Table 1: **Results on object discovery.** Two metrics, mean Intersection over Union (mIoU) and Adjusted Rand Index (ARI) are reported in % (mean ± std for 3 trials) on CLEVR6.

|  | mIoU (↑) | ARI (↑) |
|---|---|---|
| Previous methods | | |
| SA (Locatello et al., 2020) | 47.3 ± 23.2 | 63.1 ± 54.5 |
| +ARK (Kim et al., 2023) | **68.8 ± 0.4** | **95.4 ± 0.5** |
| Ours (SlotAug) | | |
| Base model (v1) | **68.9 ± 0.1** | **95.7 ± 0.2** |
| + AIM (v2) | 68.5 ± 0.1 | 95.3 ± 0.1 |
| + AIM + SCLoss (v3) | 68.5 ± 0.1 | **95.2 ± 0.7** |

Table 2: **Results on durability test** with MSE on CLEVR6.

|  | Slot (↓) | Obj. Pos. (↓) |
|---|---|---|
| | Single step (x8) | |
| v1 | 50.8 | 0.14 |
| v2 | 39.7 | 0.15 |
| v3 | **0.25** | **0.01** |
| | Multiple steps (x4) | |
| v1 | 54.0 | 0.16 |
| v2 | 41.4 | 0.11 |
| v3 | **0.31** | **0.02** |

controllability is accomplished with a neglectable compromise of the performance on both the object discovery and image reconstruction tasks, as demonstrated in Tab. 1.

One may wonder how our model can excel in controlling individual slots while being trained solely on image-level manipulation without any explicit object-level supervision. We attribute this successful transition from image augmentation to object manipulation to the slots' ability to focus their attention effectively on each distinct object. This capability is achieved by the following key factors. Firstly, using slot-wise decoder (Locatello et al., 2020) enables independent decoding for each slot, eliminating dependencies on other slots. Secondly, using Attention Refining Kernel (ARK) (Kim et al., 2023) allows our model to efficiently discover individual objects without any concerns of attention leakage. These factors collectively ensure that the slots remain directed toward their corresponding objects, thereby facilitating precise object-level manipulations. We claim that these factors enable our model to seamlessly extend the knowledge learned from image-level augmentation to object-level manipulation. More discussions including theoretical proof and empirical results are provided in the Appendix to substantiate our claim.

### 4.1.2 CONDITIONAL IMAGE COMPOSITION

We introduce *conditional image composition*, an advanced version of a downstream task called compositional generation (Singh et al., 2021). From compositional generation, or image composition, we can evaluate the reusability and robustness of slots obtained from different images. As shown in Fig. 3(b), our task aims to generate novel images by not only combining but also manipulating slots collected from various images.

Owing to the direct controllability, our model is capable of rendering objects along with human intention by modifying the objects in accordance with instructions that contain values for the desired change. As shown in the row of Fig. 3(b), the number of slots for image composition (9 objects) can be expanded beyond the quantity for which the model was originally trained (up to 6 objects). We attribute this to the characteristics of slots in Slot Attention. Moreover, our model also can resolve the conflicts among multiple images regarding the relative position (or depth) of the objects as illustrated in the object 3, 5 and 8 in the third row in Fig. 3(b). From these observations, we claim that the proposed method can effectively manipulate and combine slots without sacrificing the original nature and robustness.

### 4.2 SUSTAINABILITY IN OBJECT REPRESENTATION

### 4.2.1 ITERATIVE MANIPULATION

Fig. 4 shows the results of iterative manipulation applied to a specific object along with a series of instructions including "Stay" referring to $insts_{identity}$. We can observe that all our models succeed in manipulating the target object, demonstrating that our proposed training scheme works properly. Nevertheless, it is also clear that models v1 and v2 fail to follow the instructions along the consecutive manipulations. In the case of v1, object appearances deteriorate with the emergence of abnormal artifacts from the third round. Whereas, in v2, although the collapsing issue is mitigated, an out-of-interest property, color, changes despite no instruction for such modification. These unexpected results are also triggered by the "Stay" instruction, which is intended to maintain the current state

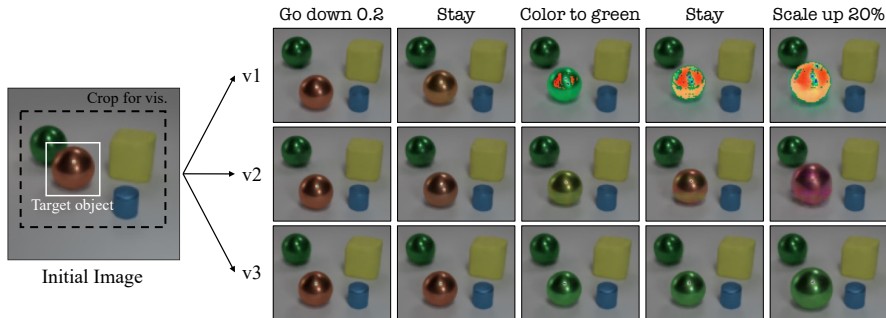

Figure 4: **Iterative slot manipulation.** The leftmost image is the initial image from which the iterative manipulation begins. The text on each column states the instruction used for manipulation. Each row shows the results of the manipulation by v1, v2, and v3 models, respectively. Center areas are cropped for better visibility.

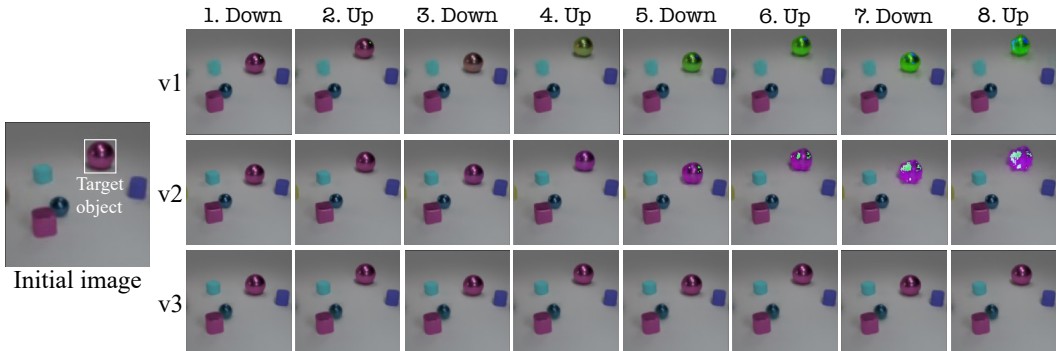

Figure 5: **Durability test.** The leftmost image is the initial image from which the test begins. The top three rows show the results of the single-step tests where each model is instructed to alternately move the target object up and down four times each. In the multi-step test, as shown in the last row, the model performs two round-trip manipulations, each involving moving the target object down, changing its color, reverting the color, and returning the object to its original position.

of the object. However, in the case of v3, we finally achieve optimal results that adhere to the instructions, including "Stay". Based on these observations, we argue that both AIM and SCLoss significantly contribute to sustainable controllability.

### 4.2.2 DURABILTIY TEST

In the durability test, we evaluate how many manipulations a model can endure while preserving object representation intact. Our durability test consists of two types: single- and multi-step tests. In the single-step test, we repeatedly manipulate slots with two instructions: one to modify a specific object property and another to revert the object to its initial state. The multi-step test involves a series of instructions to modify an object and another series to restore it to its initial state.

As depicted in Fig. 5, our findings align with the previous experiment (Sec. 4.2.1). While v1 fails to keep the color after the second round and the color gradually deviates as the rounds progress, v2 relatively preserves the color well for the fourth round. Nevertheless, from the fifth round, the texture progressively diverges from its original. Different from the v1 and v2, v3 demonstrates strong durability despite a greater number of manipulations.

We also perform quantitative evaluations on 100 randomly selected samples in CLEVR6 to measure the intrinsic deformity of slots (Tab. 2). We conduct 8 single-step and 4 multi-step round trip manipulations, both resulting in a total of 16 manipulations. We assess the durability test results by measuring the difference, using L2 distance, between the original state and the manipulated state for two aspects: the slot vector and object position vector. Both qualitative and quantitative results lead us to that our model can achieve better sustainability as the model evolves from v1 to v2 and v3.

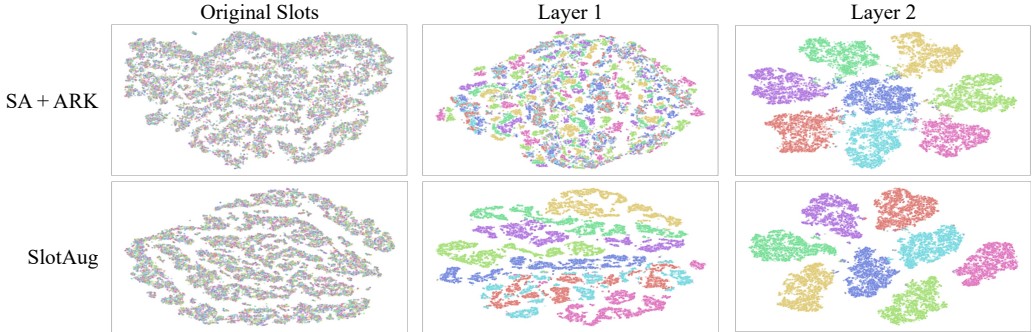

Figure 6: **t-SNE of slots on property prediction for color.** The upper row is the results for the baseline model, SA + ARK, and the lower row is the results for our method, SlotAug. The first column is the result of the original slots obtained from the spatial binding process. The second and third columns are the results of the intermediate outputs from the first and second MLP layers of the property predictor, respectively. The results of t-SNE for other properties are shown in the Appendix.

Table 3: **Results of property prediction.** Each column reports the F1 score (%) for predicting size, color, material, shape, and position, respectively. The numbers inside the parenthesis indicate the number of classes. For the position, we set two distance thresholds indicated as 'pos@threshold'.

|          | size (2) | color (8) | material (2) | shape (3) | pos@0.15 | pos@0.05 |
|----------|----------|-----------|--------------|-----------|----------|----------|
| SA + ARK | 69.7     | 63.5      | 70.4         | 59.1      | 71.1     | 51.8     |
| SlotAug  | **82.2** | **78.2**  | **82.6**     | **73.0**  | **84.2** | **77.2** |

### 4.3 SLOT SPACE ANALYSIS: PROPERTY PREDICTION

In addition to the pixel space analysis in the previous sections, we also perform slot space analysis to comprehensively assess the effectiveness of our method. We conduct *property prediction* to evaluate the quality of slots concerning human-interpretable object properties such as size, color, material, shape, and position. Through the property prediction, we can examine how well slots are distributed in the slot space along the properties of corresponding objects.

A *property predictor*, consisting of 3-layer MLPs, takes slots as input and predicts a property of objects. Each property predictor is trained by supervised learning using the ground truths. To investigate the effectiveness of object representations learned through the proposed method, the OCL models to produce slots are frozen during property prediction. As shown in Tab. 3, our model outperforms the baseline method (Kim et al., 2023) across all properties including those, like material and shape, that are not addressed during training. Moreover, in Fig. 6, qualitative results using t-SNE (Van der Maaten & Hinton, 2008) show that while the original slots themselves do not appear to be well-clustered, slots obtained by SlotAug exhibit better adaptability to the downstream task compared to those from the baseline model, reinforcing the quantitative findings. Based on these results, we assert that our method enhances interpretability not only in the pixel space but also in the slot space.

## 5 CONCLUSION

We presented an OCL framework, SlotAug, for exploring the potential of interpretable controllability in slots. To resolve the lack of labeled data, we employed image augmentation for self-supervised learning of our model. Moreover, we introduced a concept of sustainability in slots, achieved by the proposed method AIM and SCLoss. We substantiated the effectiveness of our methods by providing extensive empirical studies and theoretical evidence in the Appendix. These empirical studies include pixel- and slot-space analyses on tasks such as the durability test and property prediction. Though our work remains several questions detailed in the Appendix and represents just one step on a long journey of OCL, we firmly believe that our work is a foundational piece in the field of interpretable OCL and propel the ongoing effort to equip machines with human-like comprehension abilities.

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
