# Appendix for
# Towards Interpretable Controllability in Object-Centric Learning

## A   Limitations and Future Works

**Advanced object manipulation and real-world datasets.** In this research, we leverage image augmentation techniques to create pseudo labels for object manipulation at the image level, effectively addressing the absence of object-level ground truths. Consequently, we gain control over objects within the scope of augmentation-related properties, including attributes such as color, position, and size. Nonetheless, relying solely on image augmentation to generate supervision signals has its inherent limitations. These limitations encompass factors like the diversity of target properties and the extent of manipulation that can be effectively covered. Exploring avenues for incorporating more informative open-source datasets, such as those used for image captions, holds promise for manipulating object representations more broadly. We will elaborate this potential in Section E.1.

**Position insensitive representation.** In our research, it is observed that the Slot Attention algorithm generates slots exhibiting sensitivity to the positioning of objects. Notably, this phenomenon persists even when we exclude the soft positional encoding in the visual encoder. To achieve a more interpretable object representation, exploring the generation of well-balanced slots across various properties, rather than solely focusing on position, shows potential for future work.

**State-aware slot manipulation.** In our slot manipulation process, the Property Encoder encodes each property without taking into account the current state of the manipulated slot. For instance, when modifying the size of an object, the PropertyEncoder produces a property vector, `inst_vec`, irrespective of the color or position of the target object. By incorporating the current state of the target slot into the slot manipulation process, the precision and complexity of the algorithm could be potentially enhanced.

## B   Preliminary: Spatial Binding in Slot Attention

---

**Algorithm A** Spatial binding in slot attention algorithm in pseudo-code format. The input image is encoded into a set of $N$ vectors of dimension $D_{input}$ which is mapped to a set of $K$ vectors with dimension $D_{slot}$. Slots are initialized from a Gaussian distribution with learned parameters $\mu, \sigma \in \mathbb{R}^{D_{slot}}$. The number of iterations is set to $T = 3$.

---

```
 1: function SPATIALBINDING(img ∈ ℝ^{H×W×3})
 2:     inputs = Encoder(img)
 3:     inputs = LayerNorm(inputs)
 4:     for t = 0 … T do
 5:         slots_prev = slots
 6:         slots = LayerNorm(slots)
 7:         attn = Softmax(1/√D_slot k(inputs)·q(slots)^T, axis='slots')
 8:         updates = WeightedMean(weights=attn+ε, values=v(inputs))
 9:         slots = GRU(state=slots_prev, inputs=updates)
10:         slots = slots + MLP(LayerNorm(slots))
11:     end for
12:     return slots
13: end function
```

---

The core mechanism of the slot attention, the spatial binding, is described in Alg. A. Given an input image $\texttt{img} \in \mathbb{R}^{H \times W \times 3}$, convolutional neural networks (CNNs) encoder generates a visual feature map $\texttt{input} \in \mathbb{R}^{N \times D_{enc}}$, where $H$, $W$, $N$, and $D_{\text{enc}}$ are the height and width of the input image, the number of pixels in the input image ($= HW$), and the channel of the visual feature map. The slot attention module takes $\texttt{slots}$ and $\texttt{inputs}$, and projects them to dimension $D_{slot}$ through linear transformations $k$, $q$, and $v$. Dot-product attention is applied to generate an attention map, $\texttt{attn}$, with query-wise normalized coefficients, enabling slots to compete for the most relevant pixels of the visual feature map. The attention map coefficients weight the projected visual feature map to produce updated slots, $\texttt{updates}$. With the iterative mechanism of the slot attention module, the slots can gradually refine their representations.

## C  IMPLEMENTATION AND EXPERIMENTAL DETAILS

### C.1  TRAINING

We use a single V100 GPU with 16GB of RAM with 1000 epochs and a batch size of 64. The training takes approximately 65 hours (wall-clock time) using 12GB of RAM for the CLEVR6 dataset, and 22 hours using 9GB of RAM for the Tetrominoes dataset, both with 16-bit precision.

### C.2  IMAGE AUGMENTATION

Upon receiving an input image $img_{input}$, we produce four outputs: a reference image, denoted as $img_{ref}$, an augmented image, represented as $img_{aug}$, and the transformation instructions between them, indicated as $insts_{ref2aug}$ and $insts_{aug2ref}$.

In the data augmentation process, three pivotal variables are defined. The first is the template size $\mathcal{T}$, employed for the initial cropping of $img_{input}$ prior to the application of transformation (240 for CLEVR6 and 80 for Tetrominoes). Next, the crop size $\mathcal{C}$ is used to crop the transformed image before resizing it to $\mathcal{M}$ (192 for CLEVR6 and 64 for Tetrominoes). This two-stage cropping procedure mitigates the zero-padding that results from transformations. Lastly, the image size $\mathcal{M}$ denotes the final image size post data augmentation (128 for CLEVR6 and 64 for Tetrominoes).

In the training phase, $img_{ref}$ is obtained by applying a center-crop operation on $img_{input}$ using $\mathcal{C}$ and then resizing it to $\mathcal{M}$. The generation of $img_{aug}$ is more complex, entailing the application of a random transformation from a set of three potential transformations. Initially, $img_{input}$ is cropped using $\mathcal{T}$, and the transformation process is implemented. Following this, the transformed image is cropped by $\mathcal{C}$ and then resized to $\mathcal{M}$, yielding $img_{aug}$. The detailed description for each transformation is as follows:

**Translating.** We set a maximum translation value $d_{max} = \frac{\mathcal{T}-\mathcal{C}}{2}$. A value is randomly chosen within the range of $(-d_{max}, d_{max})$ for translation along the $x$-axis ($d_x$) and the $y$-axis ($d_y$) respectively.

**Scaling.** The maximum and minimum scaling factors, $s_{max}$ and $s_{min}$, are computed by $\frac{\mathcal{T}}{\mathcal{C}}$ and $\frac{\mathcal{C}}{\mathcal{T}}$, respectively. A float value $s$, serving as a scaling factor, is then randomly sampled from within the range of $(s_{max}, s_{min})$. One thing to note is that calculating the transformation instructions is not straightforward due to the potential translation of objects during scaling. Thus, to calibrate the instructions, we infer translation values from the predicted object positions before scaling. The position prediction is calculated as the weighted mean on the attention maps between the visual encodings and slots. With this position prediction, we add the translation term into the scaling process so that the model should perform both object-level scaling and translating: $\vec{d} = (s - 1)(\vec{p} - \vec{c})$, where $\vec{d}$ represents the vector of the translation value, $\vec{p}$ refers to the vector of the predicted object position, and $\vec{c}$ is the vector corresponding to the position of image center.

**Color shifting.** In this study, we employ the HSL (hue, saturation, and lightness) color space for effective object color manipulation. The input image, initially in RGB space, is converted to HSL space. We adjust the hue by rotating it using randomly sampled angles that span the entire hue space. For saturation, we apply a scaling factor, determined by the exponential of a value randomly drawn from (-1, 1), a hyper parameter. Our primary focus lies on the internal color of objects, leaving lightness untouched. Nonetheless, adjustments to lightness can be made if necessary.

**Instruction.** Each transformation instruction is a list of 6 values: one scaling factor ($\Lambda$scale), two translation parameters ($\Delta x, \Delta y$), and three color shifting parameters in HSL ($\Delta$hue, $\Lambda$saturation, $\Lambda$lightness) where $\Lambda$ and $\Delta$ means the multiplicative and additive factor for the corresponding values, respectively. The identity instruction, $insts_{identity}$, contains the base values for each transformation. Thus, $insts_{identity}$ has 1 for scaling, $(0, 0)$ for translation, and $(0, 1, 1)$ for color shifting. For the inverse instruction, $insts_{aug2ref}$ has the values of $-insts_{ref2aug}$ for additive factors, and $\frac{1}{insts_{ref2aug}}$ for multiplicative factors.

## C.3 MODEL

Basically, our model framework is built on Slot Attention (Locatello et al., 2020), thereby the encoder, decoder, and slot attention module are the same as that of Slot Attention except for the inclusion of the Attention Refining Kernel (ARK) from SLASH (Kim et al., 2023). For Tetorminoes and CLEVR, we employ a 4-layer CNN encoder and a 6-layer Spatial Broadcast (SB) decoder (Watters et al., 2019) with a hidden dimension of 64. Within the slot attention module, we set the slot dimension to 64, perform the binding process for 3 iterations, and use a kernel size of 5 for the ARK. Please refer to the original papers (Kim et al., 2023; Locatello et al., 2020) for additional details for Slot Attention.

For CLEVRTEX6 and PTR datasets which include more complicated objects, we adopt larger models with a slot dimension, $D_{slot}$, of 256. As encoders, we use 1) Resnet34 (He et al., 2016) following (Elsayed et al., 2022; Biza et al., 2023) and 2) ViT-base (Dosovitskiy et al., 2020), with the patch size of 8, pretrained via MAE (He et al., 2022) As decoders, we use an increased size of SB decoder consisting of 8-layer CNNs with a hidden dimension of 128, and a Transformer-based decoder proposed in SRT (Sajjadi et al., 2022b). The original SRT decoder is designed to operate at the image level, and the following research OSRT (Sajjadi et al., 2022a) introduce a modification to decode slots simultaneously. In this paper, we slightly modified it to decode each slot independently following the spatial broadcast decoder. This selection is made to demonstrate that our proposed method is not limited to CNN-based spatial broadcast decoders used in Slot Attention but can robustly operate within transformer-based decoders as well, given the appropriate conditions for independence. The results of using large models are described in Sec. E.

In Alg. 2 of the main paper, the Property Encoder (`PropertyEncoder`) takes as input the values that correspond to specific properties. Accordingly, the input size for the property encoder is 1 for scaling, 2 for translation, and 3 for color shifting. Each property is encoded via Property Encoder, a 3-layer MLP with ReLU activation functions, resulting in a `inst_vec`, a vector of dimension $D_{slot}$.

## D FROM THE IMAGE-LEVEL TRAINING TO OBJECT-LEVEL INFERENCE

To begin with, we would like to highlight our unique approach to the training procedure. While our training incorporates manipulations at the *image-level*, it can be perceived as training the model at the individual *object-level*. In this section, we discuss on how this transition is achieved without the need for an additional tuning process, and present empirical results that support our claim.

As we discussed shortly in Sec. 3.1. in the main paper, the success of transitioning from image-level augmentation during training to object-level manipulation during inference can be attributed primarily to the fact that the entire process for each slot, including object discovery and decoding, exclusively influences the reconstruction of its respective *object*. To substantiate our claim, a mathematical proof is provided below to show how an image-level reconstruction loss can be disentangled into object-level reconstruction losses.

$$\mathcal{L}_{\text{recon}} = \|\hat{\mathcal{I}} - \mathcal{I}\|_2^2 \tag{1}$$

$$= \|\sum_{k=1}^{\mathcal{K}}(\hat{\mathcal{I}}_k^{rgb} \odot \hat{\mathcal{I}}_k^{\alpha}) - \mathcal{I}\|_2^2 \tag{2}$$

$$= \|\sum_{k=1}^{\mathcal{K}}(\hat{\mathcal{I}}_k^{rgb} \odot \hat{\mathcal{I}}_k^{\alpha}) - \sum_{k=1}^{\mathcal{K}}(\mathcal{I} \odot \hat{\mathcal{I}}_k^{\alpha})\|_2^2 \tag{3}$$

$$= \| \sum_{k=1}^{\mathcal{K}} (\hat{\mathcal{I}}_k^{rgb} \odot \hat{\mathcal{I}}_k^{\alpha} - \mathcal{I} \odot \hat{\mathcal{I}}_k^{\alpha}) \|_2^2 \tag{4}$$

$$\approx \| \sum_{k=1}^{\mathcal{K}} (\hat{\mathcal{O}}_k - \mathcal{O}_k) \|_2^2, \tag{5}$$

$$= \sum_{k=1}^{\mathcal{K}} \| (\hat{\mathcal{O}}_k - \mathcal{O}_k) \|_2^2 + \sum_{\substack{i,j=1 \\ i \neq j}}^{\mathcal{K}} (\hat{\mathcal{O}}_i \cdot \hat{\mathcal{O}}_j - 2\,\hat{\mathcal{O}}_i \cdot \mathcal{O}_j + \mathcal{O}_i \cdot \mathcal{O}_j) \tag{6}$$

$$\approx \sum_{k=1}^{\mathcal{K}} \| (\hat{\mathcal{O}}_k - \mathcal{O}_k) \|_2^2, \tag{7}$$

where $\mathcal{K}$ is the number of slots, $\hat{\mathcal{I}} \in \mathbb{R}^{H \times W \times 3}$ represents the reconstructed image, and $\mathcal{I} \in \mathbb{R}^{H \times W \times 3}$ represents the input image. $\hat{\mathcal{I}}_k^{rgb} \in \mathbb{R}^{H \times W \times 3}$ and $\hat{\mathcal{I}}_k^{\alpha} \in \mathbb{R}^{H \times W \times 1}$ are the reconstruction results generated by the decoder using the k-th slot as input: an RGB and an alpha map (or an attention mask), respectively. $\hat{\mathcal{O}}_k \in \mathbb{R}^{H \times W \times 3}$ is the predicted image for the specific object that is bounded with the k-th slot, while $\mathcal{O}_k \in \mathbb{R}^{H \times W \times 3}$ is the corresponding ground-truth object image.

From Eq. (1) to Eq. (2), we follow the decoding process of Slot Attention (Locatello et al., 2020). In particular, each k-th slot is decoded independently, resulting in the reconstructed RGB image $\hat{\mathcal{I}}_k^{rgb}$ and the reconstructed alpha map $\hat{\mathcal{I}}_k^{\alpha}$. The final reconstruction image $\hat{\mathcal{I}}$ is generated by aggregating $\hat{\mathcal{I}}_k^{rgb}$ using a pixel-level weighted average, where the weights are determined by $\hat{\mathcal{I}}_k^{\alpha}$. It is crucial to recognize that $\hat{\mathcal{I}}_k^{\alpha}$ serves as an attention mask, as elaborated below:

$$\sum_{k=1}^{\mathcal{K}} \hat{\mathcal{I}}_k^{\alpha}(x, y) = 1 \quad \text{for all } x, y, \tag{8}$$

where $\hat{\mathcal{I}}_k^{\alpha}(x, y)$ is a value for the position $(x, y)$. This characteristic plays a pivotal role in our approach, facilitating the transition from Eq. (2) to Eq. (3). In this transformation, the input image $\mathcal{I}$ is effectively weighted by the set of $\mathcal{K}$ alpha maps, denoted as $\hat{\mathcal{I}}_k^{\alpha}$, where $k$ spans from 1 to $\mathcal{K}$. Then, as both the first and second terms in Eq. (3) involve the same sigma operations, we can simplify the expression by combining the individual subtraction operations into a single sigma operation (Eq. (4)).

Subsequently, we approximate Eq. (4) as Eq. (5) to get an object-level disentangled version of the reconstruction loss. Here we assume that both $\hat{\mathcal{O}}_k$ and $\mathcal{O}_k$ only consist of a specific region of interest within the input image. This region corresponds to the target object which is bound to the $k$-th slot, while the remaining areas are masked out and assigned a value of zero. We can make this assumption based on the successful performance of the previous object-centric learning model, SLASH (Kim et al., 2023). SLASH has demonstrated effective capabilities in focusing on and capturing specific objects of interest within an image, by introducing the Attention Refining Kernel (ARK). By incorporating ARK into our model, we confidently assume that $\hat{\mathcal{O}}_k$ and $\mathcal{O}_k$ primarily represent the target object while masking out other irrelevant parts as zero as shown in Fig. A.

Here, we would like to note that ARK is an optional component in our method, not a necessity. The use of ARK is not intended to enhance object discovery performance in a single training session; rather, it is employed to ensure consistent results across multiple experiments. If our proposed training scenario arises where bleeding issues do not occur in the original SA, it can be achieved without the need for ARK. To substantiate this claim, we present qualitative results in Fig. C, where we train SlotAug with the original SA (without ARK). One can easily catch that the object manipulation fails in the case of bleeding problem. Specifically, the analysis for the failure case in bleeding problem is as follows: 1) Obviously, if the attention map corresponding to the target object encompasses other objects, it becomes impossible to exclusively manipulate solely the target object, leading to unexpected artifacts in other objects. 2) Whenever tinting instructions are applied, objects become

gray and we attribute this to the backgrounds – having a gray color – intervening with the target objects during training.

Eq. (5) can be broken down into two separate summations. The first one is our target term that is the sum of object-level MSE losses, and the second term is the residual term. Lastly, the transition from Eq. (6) to Eq. (7) constitutes a significant simplification in the representation of the loss function. This is a valid transformation under the assumption follows:

$$\hat{\mathcal{O}}_i \cdot \hat{\mathcal{O}}_j = \hat{\mathcal{O}}_i \cdot \mathcal{O}_j = \mathcal{O}_i \cdot \mathcal{O}_j = 0 \quad \text{if } i \neq j. \tag{9}$$

This assumption postulates that the inner product of different object images, whether they are predicted or ground-truth, is always zero. We assert that this assumption is justifiable, much like the previous one, given the promising results obtained in our object discovery experiments. The loss computation is thus decomposed into individual components for each slot, which lends itself to an interpretation of object-level loss.

The conversion from image-level MSE loss to a sum of individual object-level MSE losses provides a new perspective on our training method. Despite the use of image-level manipulations, the underlying core of the training process inherently engages with object-level representations. This demonstrates how a simple methodological addition, incorporating image augmentation into the training process, can lead to considerable gains in the model's capacity for user-intention-based object manipulation.

Fig. A empirically demonstrates the effectiveness of our model, leveraging Slot Attention for controllability over slots. Conversely, it was noted that the well-known alternative framework for object-centric learning, SLATE (Singh et al., 2021), employing image tokenization from Discrete VAE (dVAE) (Im Im et al., 2017) and Transformer-based auto-regressive decoding (Vaswani et al., 2017), struggled with the manipulation of slots, as illustrated in Fig. B. The same slot manipulation strategy via Property Encoder was used for comparison. Other training environments are just the same as the official paper (Singh et al., 2021) except for the addition of the training loss for the reconstruction of the augmented images.

## E FURTHER EXPERIMENTAL RESULTS

In this section, we present more qualitative results including failure cases (Fig. D), and object-level manipulation using various backbone encoders and decoders (Fig. E and F). Furthermore, we showcase several additional experimental results below.

### E.1 FULLY SUPERVISED TRAINING ON MATERIALS AND SHAPES

To explore the capabilities of our method when provided with human-annotated labels, we demonstrate object manipulation examples related to materials and shapes. We utilize the CLEVR render [1] to generate datasets having ground truth in terms of property modification. Using datasets containing precise annotations for the target properties (materials and shapes), we can explicitly train a model through object-level supervision.

As shown in Fig. G, the model can effectively acquire knowledge of extrinsic properties, such as material and shape, when provided with appropriate supervision signals. In future work, one can aim to enhance the proposed training scheme by leveraging more informative datasets, such as those for image captioning (Chen et al., 2015; Wang et al., 2023), to train a more human-interactive framework. This process may entail elaborate data processing since the datasets were not originally designed for the purposes of object manipulation. However, we firmly believe that pursuing this path holds great promise.

### E.2 QUANTITATIVE EVALUATION ON OBJECT MANIPULATION

The supplementary quantitative results on object manipulation are shown in Tab. A. Given the absence of an evaluation benchmark dataset, we opt to employ the same CLEVR render as described in Sec.

---

[1]https://github.com/facebookresearch/clevr-dataset-gen

Table A: **Results of object-level manipulation on the rendered dataset.** We evaluate the object-level manipulation by assessing metric scores over three sorts of images: reference images (ref.); manipulated images (manip.); and reversion or restored images (rev.). We use mIoU and ARI for the object discovery task, and MSE for the image generation task.

|    | mIoU | | | ARI | | | MSE | | |
|----|------|--------|------|------|--------|------|--------|--------|--------|
|    | ref. | manip. | rev. | ref. | manip. | rev. | ref. | manip. | rev. |
| v1 | 89.4 | 69.9 | 68.9 | 97.9 | 79.3 | 80.1 | 7.3e-4 | 4.6e-3 | 6.3e-3 |
| v2 | 87.9 | 71.4 | 80.0 | 96.1 | 79.4 | 90.2 | 7.5e-4 | 4.2e-3 | 3.1e-3 |
| v3 | 85.4 | 70.5 | 79.8 | 95.3 | 78.8 | 89.8 | 7.8e-4 | 3.5e-3 | 2.1e-3 |

Table B: **Results of ablation studies on weights for the training loss.** The leftmost column shows the values of the weight for SC-Loss ($w_{cycle}$), while the weight for the reconstruction loss ($w_{recon}$) is set to $1.0$. The other columns show the training losses when training is finished and the scores of the validation metrics for the object discovery task. Each row shows the results of using the weight for SC-Loss with $1.0$, $0.1$, and $0.01$, respectively.

| $w_{cycle}$ | Train | | | Val | |
|-------------|---------------|----------------|------------|------|------|
|             | loss_recon_ref | loss_recon_aug | loss_cycle | mIoU | ARI |
| 1.0*  | 3.5e-4 | 4.9e-4 | 2.2e-6 | 66.5 | 94.1 |
| 0.1   | 3.2e-4 | 3.8e-4 | 1.8e-5 | 68.5 | 95.2 |
| 0.01  | 3.1e-4 | 4.4e-4 | 6.2e-5 | 68.7 | 95.4 |

E.1, to generate a set of 1500 triplets. These triplets consist of a reference image, instructions for object manipulation, and the resulting manipulated image. Additionally, it is worth noting that there exists no prior research specifically addressing slot manipulation through direct human-interpretable instructions. Consequently, our performance comparisons are restricted to different versions of our model: v1 (the base model with image augmentation only), v2 (image augmentation + AIM), and v3 (image augmentation + AIM + SCLoss).

One can observe that all three models successfully execute object discovery and object manipulation tasks with minimal differences in their performance scores, highlighting the effectiveness of our training approach leveraging image augmentation. However, in the context of the reversion task, wherein the models are instructed to revert the manipulated objects to their original state, both v2 and v3 outperform v1, demonstrating the effectiveness of the proposed AIM. Furthermore, in terms of image editing, v3 surpasses both v1 and v2 by a significant margin, underscoring the effectiveness of the proposed SCLoss.

### E.3    ABLATION STUDY ON LOSS WEIHGTS

We conduct an ablation study on training losses using the v3 model (image augmentation + AIM + SCLoss). Tab. B shows the results of training models with three different loss weights for the SCLoss ($w_{cycle}$) while maintaining the $w_{recon}$ as 1.0. For the balanced training result, considering both image reconstruction and object discovery, we opted for 0.1 due to its balanced performance.

### E.4    EXTREME DURABILITY TEST

We evaluate our v3 model with two stringent versions of the durability test. Fig. H displays the first extreme durability test, wherein we manipulate all objects within a given scene across a total of 24 manipulation steps. The complete manipulation process encompasses four cycles of round-trip manipulations, each cycle comprising three sequential forward manipulations followed by three recovery manipulations. Despite the fact that the appearance of each object tends to deviate from its initial state as manipulations accumulate, it is notable that our model demonstrates substantial robustness against multiple rounds of manipulations.

In the second durability test, we manipulate a target object through 50 steps of manipulation, which consists of 25 cycles of a single forward manipulation (translation, scaling, or color shifting) and its

corresponding recovery manipulation. Fig. I demonstrates our model's significant endurance against numerous manipulation steps. We observe that our model exhibits greater robustness in translating objects compared to scaling and color shifting. However, it is noteworthy that the resilience of our model against both scaling and color shifting is impressive, as it withstands around 20 steps of manipulations without significant distortion in object appearance. The figure further includes qualitative results that gauge slot divergence. It becomes clear that our model's durability improves gradually as it evolves from version v1 to v2 and finally to v3.

### E.5    TOY APPLICATION: OBJECT-CENTRIC IMAGE RETRIEVAL

With the acquisition of object-level controllability, we can extend object-centric learning to a newly introduced downstream task, called object-centric image retrieval. Object-centric image retrieval aims to retrieve an image having an object that is most relevant to a target object that is given by the user's intention.

The retrieval process is as follows. First, we acquire slots for a target object and candidate objects from the corresponding images by conducting object discovery. Then, to reduce the effect of spatial properties such as object position or size, we *neutralize* the slots by performing slot manipulation with the instructions for moving the objects to the central position and for scaling the objects to the unified size. After neutralization, we generate a *object image* by decoding a neutralized slot. The relevance scores between the target object image and candidate object images are computed along the given metric, specifically the MSE. Finally, object-centric image retrieval can be accomplished by finding the image containing the top-k objects as shown in Fig. J.

### E.6    ADDITIONAL T-SNE RESULTS

Additional t-SNE results from the property prediction are shown in Fig. **??**. Similar to the result on the color property in the main paper, we can observe that the proposed SlotAug produces more well-clustered slots in the earlier layer in the property predictors.

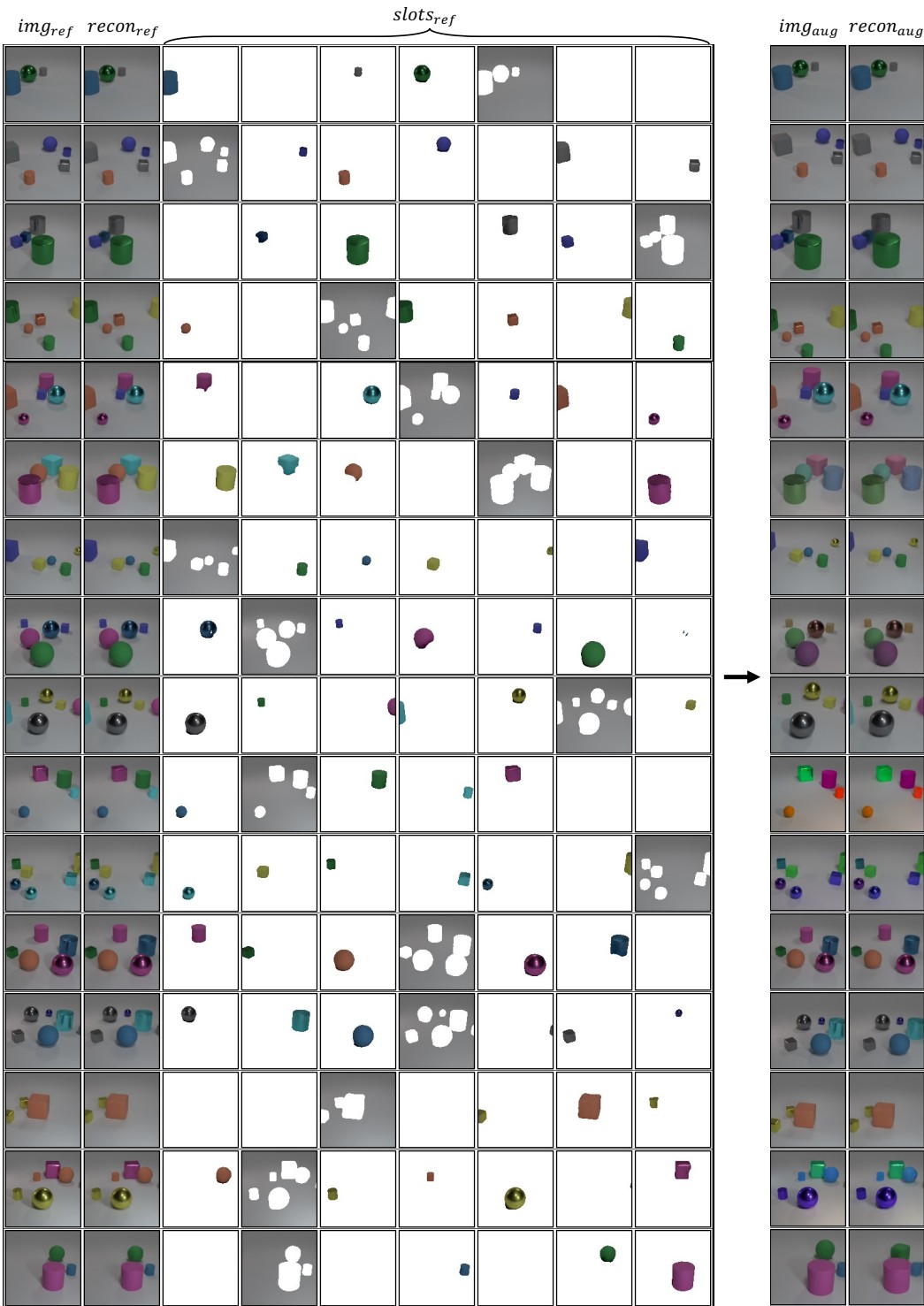

Figure A: **Training results of our method.** The leftmost column is the reference images, $img_{ref}$. The second leftmost column is the reconstruction of the reference images, $recon_{ref}$. The middle columns show the object discovery results where each column corresponds to a single slot in $slots_{ref}$. The second rightmost column is the augmented images, $img_{aug}$. The rightmost column is the reconstruction of the augmented images, $recon_{aug}$.

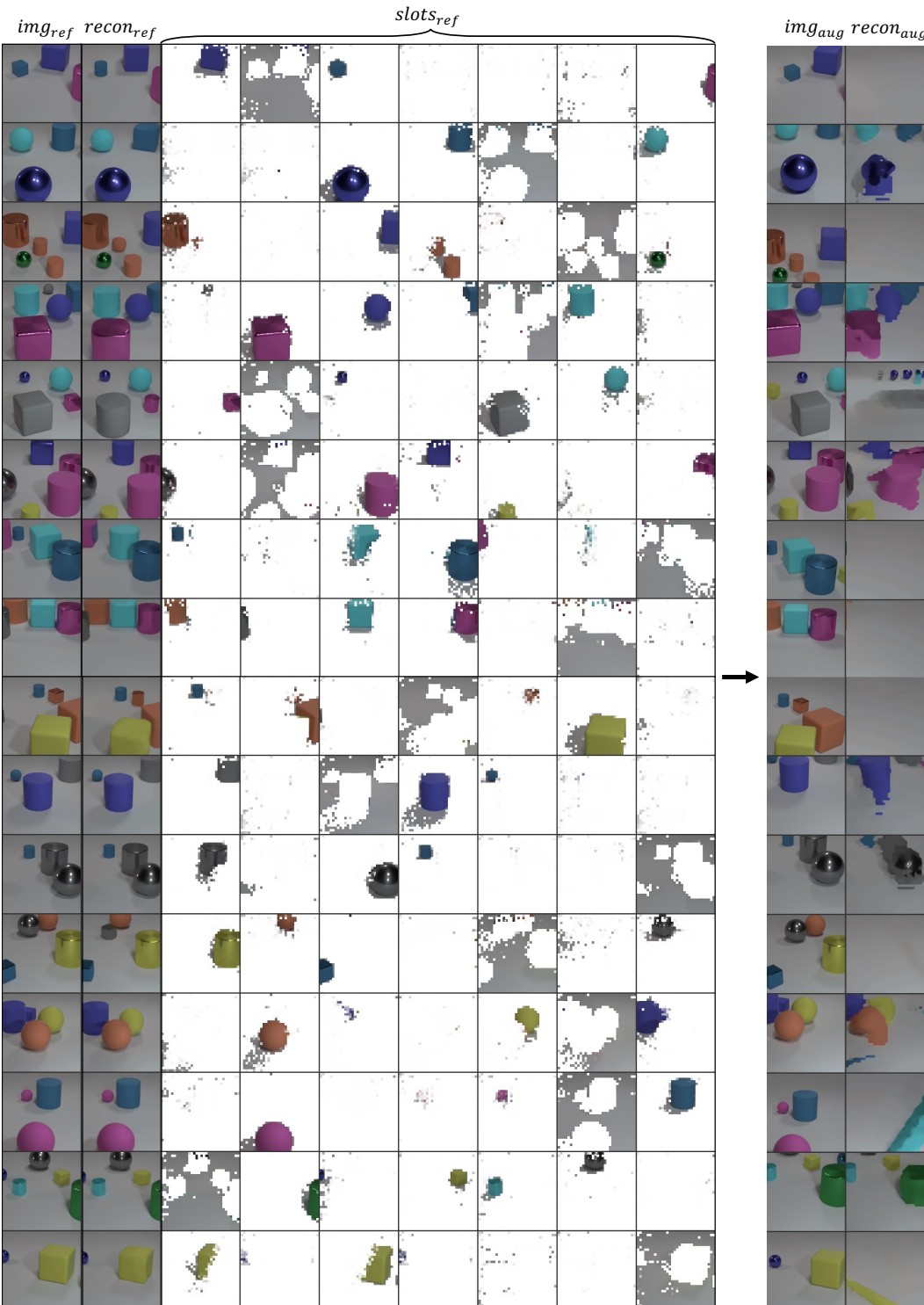

Figure B: **Training results of SLATE (Singh et al., 2021) for slot manipulation.** The leftmost column is the reference images, $img_{ref}$; The second leftmost column is the reconstruction of the reference images, $recon_{ref}$. The middle columns show the object discovery results where each column corresponds to a single slot in $slots_{ref}$. The second rightmost column is the augmented images, $img_{aug}$. The rightmost column is the reconstruction of the augmented images, $recon_{aug}$.

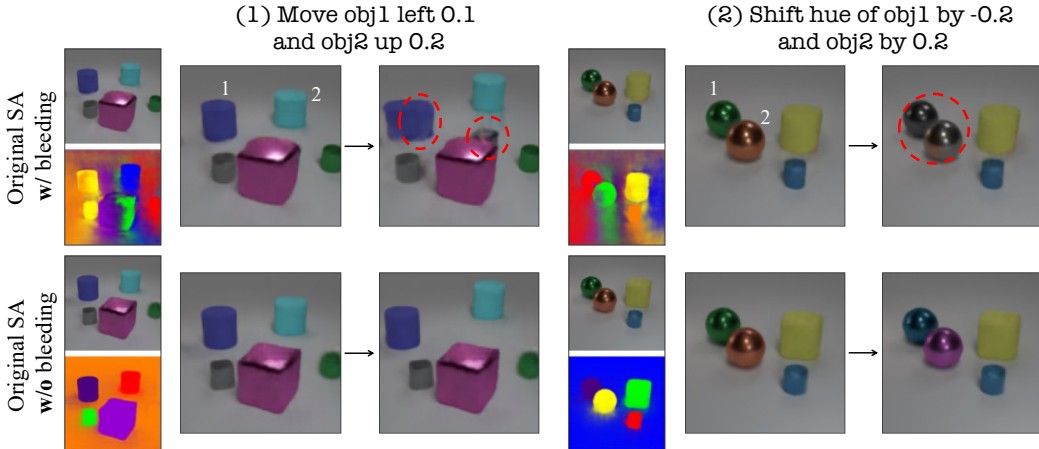

Figure C: **Visualization of object manipulation results affected by the bleeding problem with the original Slot Attention.** The first row demonstrates the cases where bleeding problem emerges, while the second row shows the cases where the object discovery is done successfully.

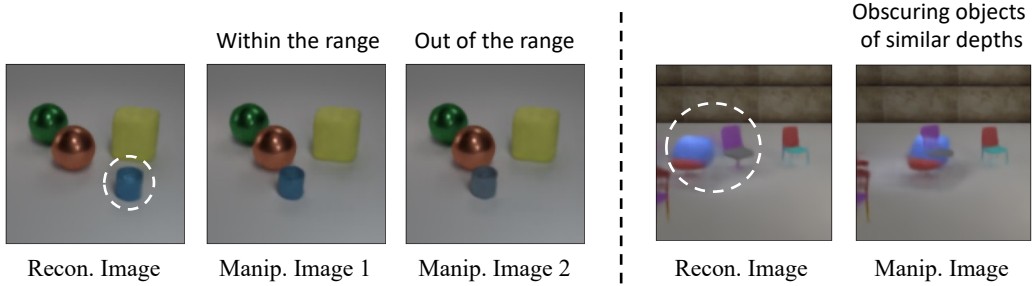

Figure D: **Visualization of failure cases.** On the left, the target object loses its original color when translated beyond the range defined in the training settings. On the right, unnatural overlapping between objects occurs when objects have similar depths.

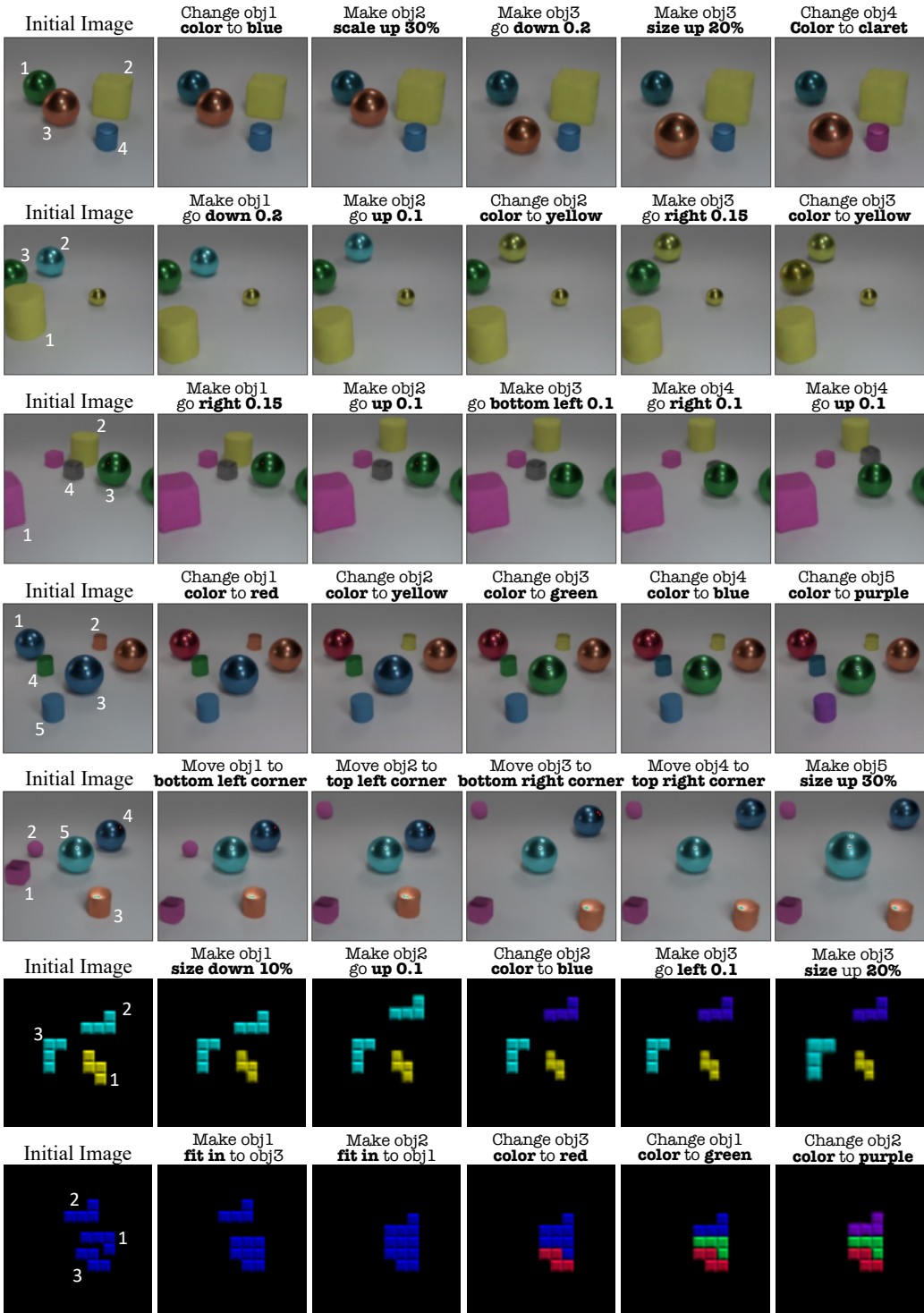

Figure E: **Visualization of object manipulation in CLEVR6 and Tetrominoes dataset.** The leftmost column features the initial images, serving as the starting point for the manipulation process. The subsequent columns depict the results of object-level manipulation, following the instructions represented as text above the images.

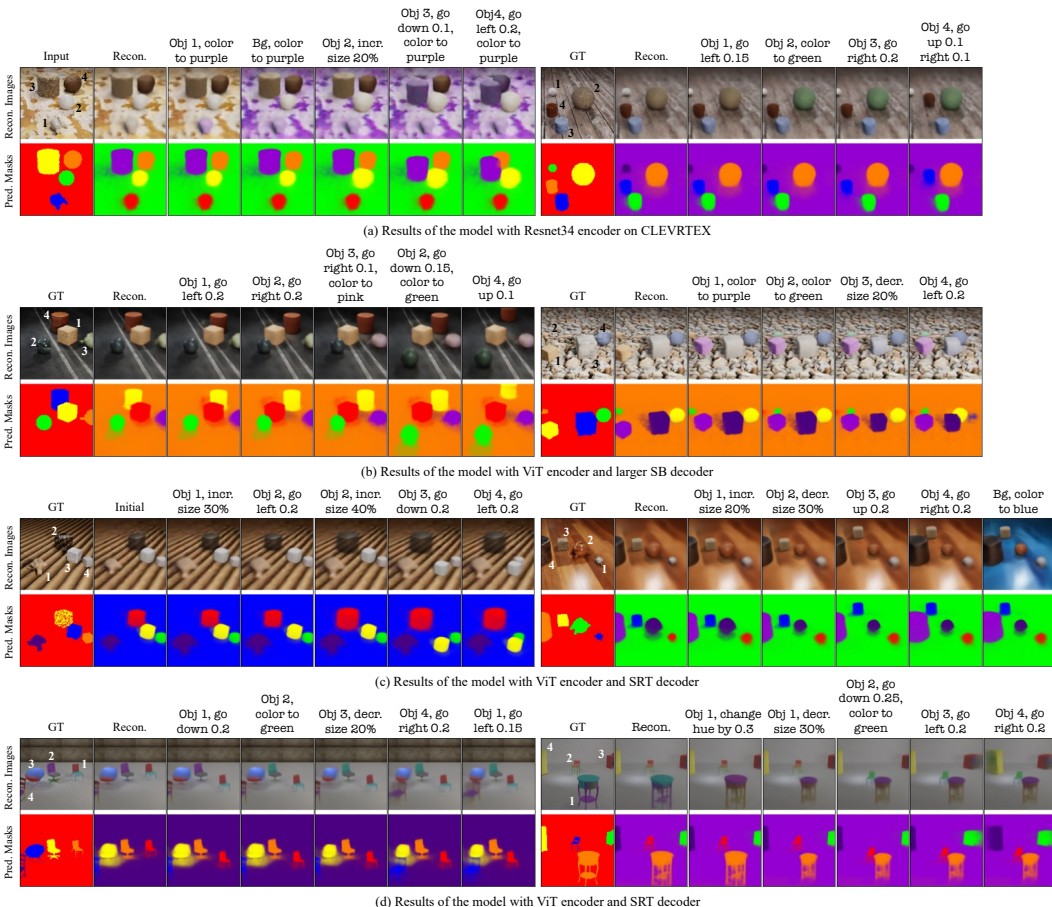

Figure F: **Visualization of object manipulation in CLEVRTEX and PTR with larger encoder and decoder.** The first three rows are for CLEVRTEX and the last is for PTR dataset. Regarding the encoder, we initialize ResNet34 randomly, while ViT is pre-trained using MAE (He et al., 2022) on the target datasets. For the decoder, we employ both the Spatial Broadcast (SB) decoder and the SRT decoder (Sajjadi et al., 2022b). In case (b), we enhance the size of the SB decoder with a hidden dimension of 128 and a depth of 8. Additionally, for SRT, we adopt a slot-wise decoding strategy akin to the SB decoder.

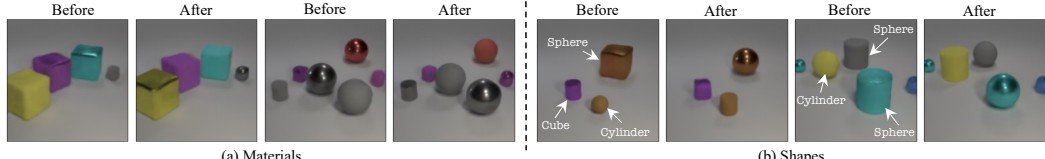

Figure G: **Results of manipulating materials and shapes of objects.** For this experiment, the training datasets are crafted using the CLEVR renderer, wherein we modify target properties such as materials and shapes while keeping other properties unchanged.

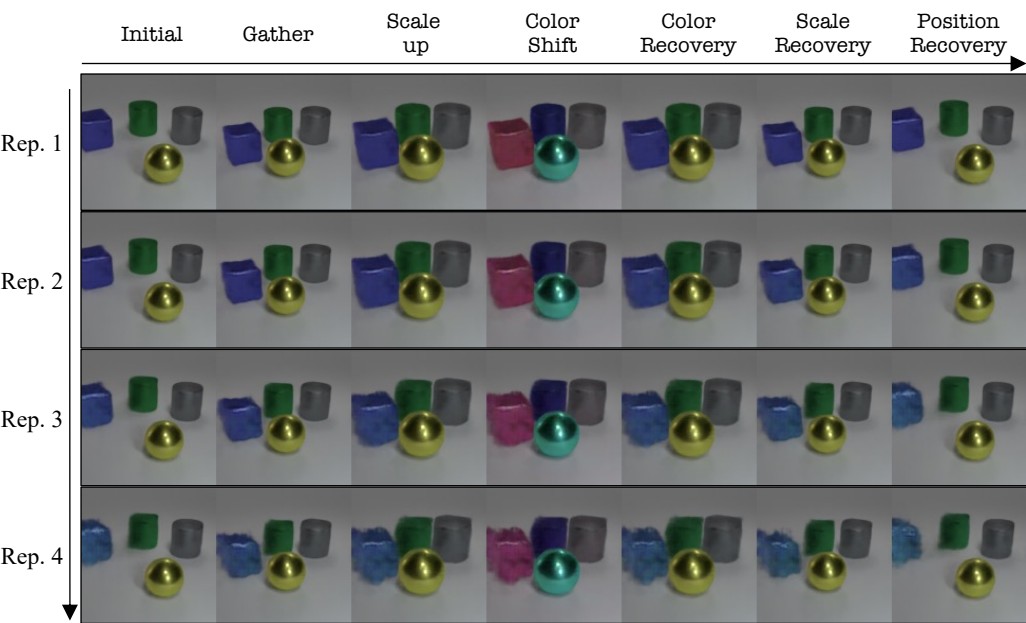

Figure H: **The results of the durability test** are depicted in the figure, wherein all objects in the scene are manipulated in accordance with the instructions, provided as text at the top of the figure. The leftmost column presents the initial state of the image. The subsequent three columns comprise three distinct manipulations: translation, scaling, and color shifting. The right three columns feature three recovery manipulations intended to restore the image to its original state. We perform 4 cycles of these round trip processes, leading to a total of 24 manipulations.

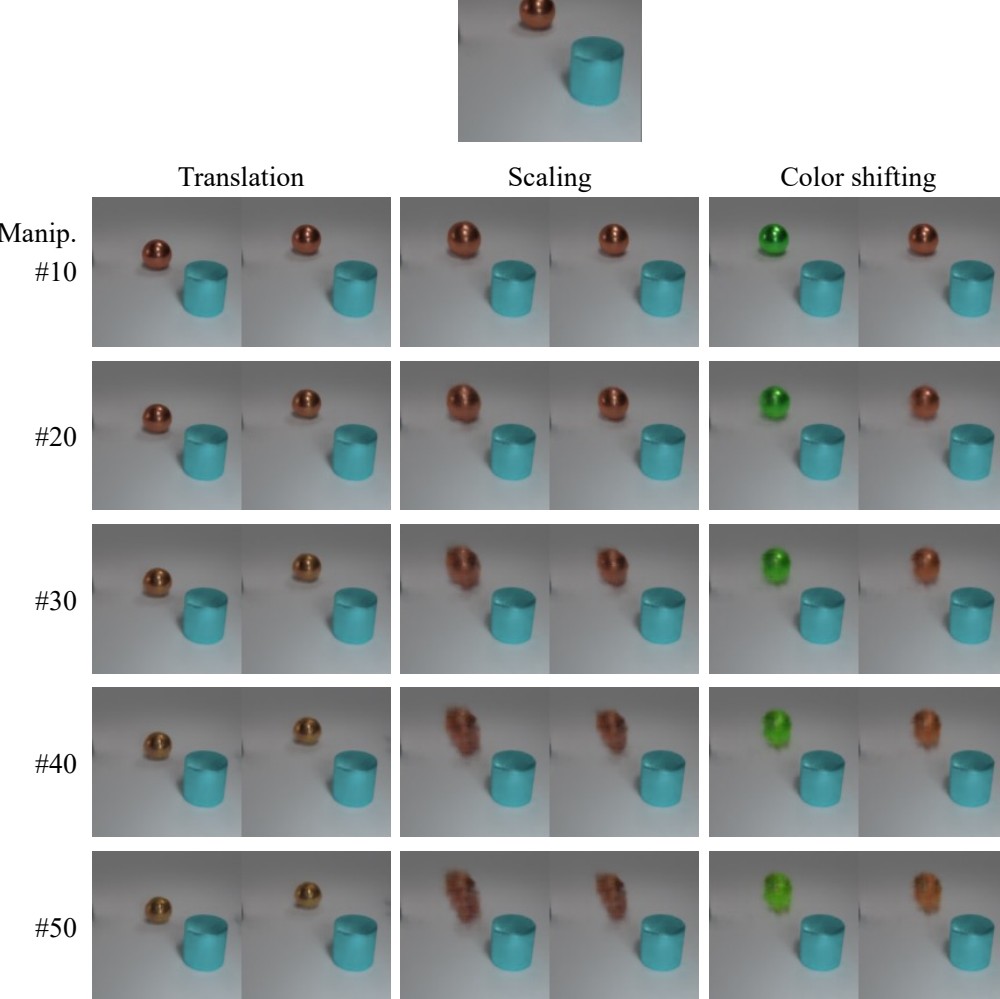

Figure I: **The results of the single-step durability test.** The top image represents the initial state prior to any manipulation. Each column depicts the results of the single-step durability test with translation, scaling, and color shifting, respectively. The left images in each column illustrate the outcome of the manipulation corresponding to the column name, while the right images in each column display the results of the recovery, or inverse, manipulation. Each row represents the results after a series of manipulations, with the number of manipulations corresponding to the row number.

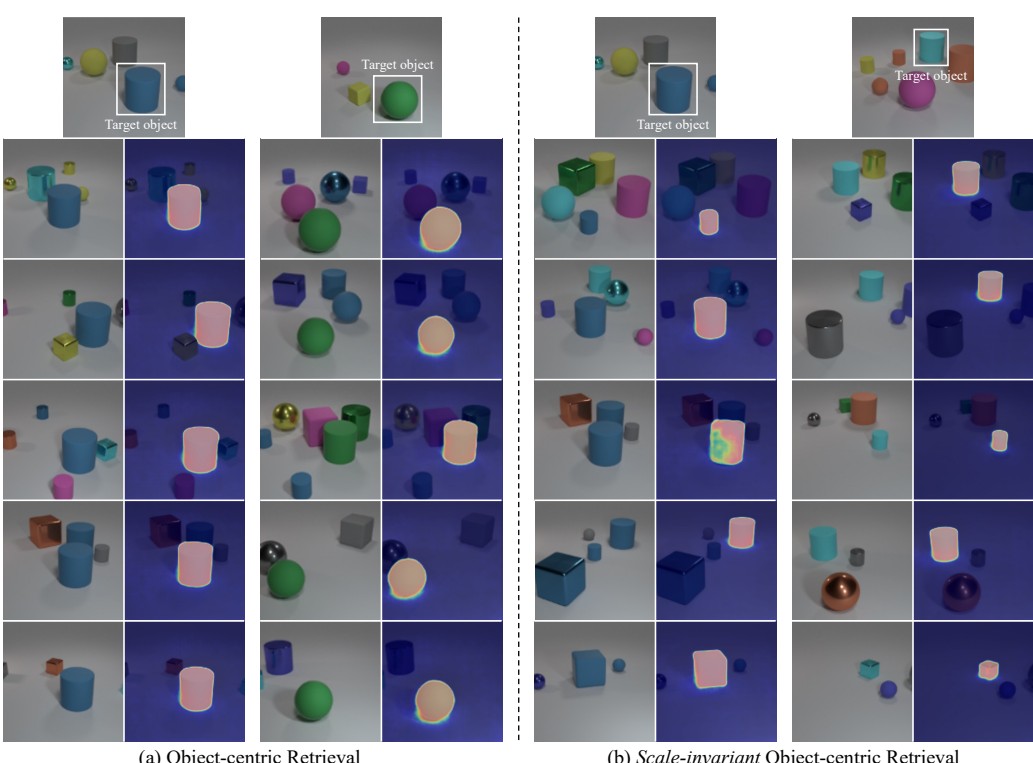

(a) Object-centric Retrieval          (b) *Scale-invariant* Object-centric Retrieval

Figure J: **Visualization of object-centric image retrieval.** The top row displays query images, indicating the target objects to be retrieved. Below each query image, you can find the top 5 retrieval results. Each retrieval result consists of the original image on the left and an attention map on the right. The attention map, associated with the slot corresponding to the target object, emphasizes the specific region within the image.