# OpenReview forum: "Towards Interpretable Controllability in Object-Centric Learning"
_ICLR.cc/2024/Conference — Submitted to ICLR 2024_

### Official Review · Reviewer_jXVc · 2023-10-29

**Soundness:** 2 fair
**Presentation:** 1 poor
**Contribution:** 2 fair
**Rating:** 1
**Confidence:** 3

**Summary:**

The paper proposes an equivariant consistency regularization for SlotAttention architectures. The idea is for each augmentation of an image, learn a corresponding augmentation that can be applied directly to the object slots, allowing users to modify object representations directly.
The paper is a bit like adding an equivariant consistency loss (e.g. https://paperswithcode.com/paper/unpaired-image-to-image-translation-using) to SlotAttention. Where image augmentations have corresponding transformations on the Slot representations.

**Strengths:**

### Related Work
- Decent coverage on learning interpretable latents using VAE/GANs

### Experiments
- I appreciate including error bars. I wish more papers did this.
- Generally well-organized experiments ssection

**Weaknesses:**

### Overall
Overall the paper is hard to follow, because common terms like equivariances are called different names (e.g. sustainability) and not clearly defined. The experiments are only on variants of CLEVR, with no evaluations on real-world data. Even on CLEVR, the results are not very convincing, and the method requires defining specific image augmentations for each equivariant action — so it is hard to use this on real-world datasets.


### References:
- Related work ignores the majority of work on object detection + localization. Learning approaches to identifying and localizing objects far predates slot attention — it’s a core computer vision task. Learning approaches go way back, too — OverFeat, deformable parts models, RCNN + children, YOLO, SAM, etc.


### Method:
The authors introduce (as a contribution) new language for agreed-upon terms like equivariance, and the new language doesn’t add anything in my opinion. It is neither intuitive nor well-defined, and only serves to make the work harder to understand.
For example:
1. “In this work, we introduce sustainability which stands for the concept that object representations should sustain their integrity even after undergoing iterative manipulations.”
What is integrity? It seems to be defined in terms of the "durability test" (AKA invertability). But equivariance and invertibility are already in common usage for a while now.


### Experiments:
Experiments are on variants of CLEVR, which is a very simple dataset that was generated in ways that privilege this algorithm. No evaluation on real-world datasets. Other work (e.g. instruct pix2pix https://arxiv.org/pdf/2211.09800.pdf) DOES show zero-shot results on real-world datasets.
Regardless, the results even on CLEVR are not convincing — leading to little to no improvement for object detection.

### Misc:
Many terms feel philosophical, when they could be stated more concretely. E.g. “Then, the model performs spatial binding on img_{ref} to produce slots{_ref}”. Meaning you run the image through the model to get the slot latents?

**Questions:**

### Comparison to related work:
For interpretable latents (e.g. VAE or GANs), the authors note that these require “manual efforts to identify the features associated with specific properties.”
    - In this work, too, you have to hand-design the augmentation and regenerate appropriate training data. This is also a manual effort, and arguably harder than a post-hoc approach?


### Definition of "Durability Test"
This is defined in the paper as “The multi-step test involves a series of instructions to modify an object and another series to restore it to its initial state.”
1. This is a fine definition, but why not also write out the equations: e.g.  $(g_1 * … * g_k)^{-1} (g_1 * … * g_k) x = x$ ?

However, the requirement that image augmentations are invertible is a strong one (e.g. viewpoint change is not invertible from the image alone). Why not make the approach more general and focus on measuring equivariance

E.g. you can measure the equivariance of the whole model by augmenting the images $(g_1 * … * g_k) (image) = (h_1 * … * h_k) (slots)$ where $g_i$ is an image aug and $h_i$ is the corresponding instruction ref2aug

---

> ### Author Response · Authors · 2023-11-17
> **Response to Reviewer jXVc (1/3)**
>
> We appreciate the time and effort the reviewer dedicated to assessing our paper.
> In our response, we aim to address the reviewer's comments comprehensively, with the hope of resolving the concerns.
>
> Before addressing specific points, we would like to offer insights into the context of our work within the domain of object-centric learning (OCL).
> This field, as a subset of compositional understanding, constitutes a significant area within computer vision, aims at achieving human-like generalization by deconstructing visual sensory inputs into symbol-like entities, such as objects.
> To achieve this, OCL focuses on generating informative object representations, called slots.
> Slightly different from other domains such as object detection and localization, which primarily focus on the final output of deep networks, OCL places its emphasis on finding intermediate representations for objects.
>
> Nevertheless, it is true, as the reviewer jXVc mentioned, that “it is hard to use this on real-world datasets” while “other works (e.g. instruct pix2pix) show zero-shot results on real-world datasets.”
> This is a prevalent problem in recent OCL research, not just ours, considering that several zero-shot and foundation models like SAM demonstrate exceptional performance across various real-world datasets in downstream tasks.
> We believe that the difference between OCL and the state-of-the-art methods for the real-world datasets can be attributed to the task-agnostic nature of OCL.
> OCL concentrates on examining how to define the objects, how to represent them, and how to understand the object interaction within the visual scene, devoid of any predefined target tasks.
> In other words, OCL research investigates meaningful patterns defining what we identify as objects, and aims to capture the objects into vector representations only with RGB images, distinct from other artificial tasks influenced by human biases.
> Therefore, in many OCL studies, the primary objective function for training is an RGB reconstruction loss in a self-supervised manner; here, the reconstruction is not even the target task of the OCL research.
>
> We firmly believe that the approach of OCL, as a stand-alone actively researched topic, holds value within computer vision and artificial neural network research for the following reasons:
> 1) OCL explores the vector representations which serves as the foundational elements of all information that we handle in the era of artificial neural network.
> 2) OCL explores the objects which are the building blocks of the visual world. Objects are self-contained, separate from one another, and interact actively with others to compose the visual world. OCL tries to interpret these objects and their relations in a vector space using object representations.
>
> Given these motivations, a lot of works [1-5] explore this domain, and below are some quotes emphasizing the significance of the OCL domain from other research papers.
> - “Learning object-centric representations of complex scenes is a promising step towards enabling efficient abstract reasoning from low-level perceptual features.” (Slot Attention [1])
> - “Object-centric representations are a promising path toward more systematic generalization by providing flexible abstractions upon which compositional world models can be built.” (SAVi [2])
> - “Learning good representations of complex visual scenes is a challenging problem for artificial intelligence that is far from solved.” (IODINE [3])
>
> In line with many valuable studies in OCL, the contribution of our work is on the first exploration of interpretable (and user-interactive) controllability over object representation — an aspect that has seen limited investigation within OCL.
> We sincerely hope that the reviewer can reconsider the view on our work within the OCL domain.
> Echoing the other reviewers’ comments, we politely claim our work is “very original” (byFg), “novel” and “encouraging for future research” (dLyp) with the simple approach as Reviewer jXVc also pointed out and Reviewer byFg mentioned as "the main advantage is its simplicity".
> With comments from other reviewers, we respectfully assert that our method is regarded as "very original" (byFg), "novel," and "encouraging for future research" (dLyp). Additionally, our approach is acknowledged for its simplicity, which Reviewer byFg also commented as "the main advantage.”
>
> [1] Locatello et al., "Object-Centric Learning with Slot Attention." (NeurIPS 2020)
>
> [2] Kipf et al., “Conditional Object-Centric Learning from Video.” (ICLR 2022)
>
> [3] Greff et al., “Multi-Object Representation Learning with Iterative Variational Inference.” (ICML 2019)
>
> [4] Burgess et al., “Unsupervised Scene Decomposition and Representation.” (arXiv 2019)
>
> [5] Singh et al., “Neural Systematic Binder.” (ICLR 2023)

---

> > ### Author Response · Authors · 2023-11-17
> > **Response to Reviewer jXVc (2/3)**
> >
> > > Terminology of sustainability (instead of equivariance)
> >
> > We agree that clarifying "sustainability" as a pursuit of "equivariance" in slot manipulation will enhance the term's understandability.
> > However, it is important to highlight that our paper presents a novel approach to slot controllability.
> > The introduction of the term "sustainability" is aimed at conceptualizing the challenges, including the durability test, in slot manipulation which has not been explored yet, while underscoring the significance of slot reusability for the robust utilizations.
> >
> > Our intention in bringing "sustainability" is to suggest a direction for OCL studies, rather than to claim novelty for the term itself.
> > We believe our novel contribution lies in the initial pursuit of interpretable controllability over slots and enhancing their robustness, with the ultimate goal of achieving equivariance in slot manipulation.
> >
> > > “The results even on CLEVR are not convincing.”
> >
> > As previously mentioned, and outlined in Section A of the appendix, we acknowledge the reviewer's observation regarding our weakness—our evaluation being limited to synthetic datasets.
> > However, we would like to emphasize that recent works in OCL [1,2,3] frequently conduct evaluations on the CLEVR dataset.
> > Furthermore, they classify CLEVRTEX and PTR as challenging datasets due to their visual complexity and intricate object patterns.
> > We conducted evaluations on these challenging datasets and believe that the results strongly demonstrate the effectiveness of our method, and this sentiment aligns with the feedback from Reviewer byFg, who mentioned, "The qualitative results are very impressive, even though the datasets are quite simple.”
> > For these reasons, we respectfully disagree with the comment — “the results even on CLEVR are not convincing” — considering the context of the OCL domain.
> >
> > > “Leading to little to no improvement for object detection.”
> >
> > That is correct.
> > The objective of our proposed method is to attain interpretable controllability over slots while minimizing any negative impact on the object discovery task.
> > Our findings, as shown in Table 1, indicate that our methods do not result in any degradation in the object discovery evaluation.
> > Hence, we kindly hope the reviewer might reconsider this aspect as a strength of our work.
> >
> > > “Meaning you run the image through the model to get the slot latents?”
> >
> > We appreciate your feedback regarding the use of certain terms in our paper.
> > The reviewer is correct in the interpretation of 'spatial binding' as the process of processing the image through the model to obtain the slot latents.
> > While we have used the term “spatial binding” in line with its technical usage in previous works [1-3], we understand the need for clarity and simplicity in our language to enhance the paper's readability.
> > Moving forward, we will strive to use more straightforward and easily comprehensible terminology.
> > We thank the reviewer for pointing out this aspect and will make the necessary revisions to address this concern.
> >
> > [1] Greff et al., "On the Binding Problem in Artificial Neural Networks." (arXiv 2020)
> >
> > [2] Singh et al., "Neural Systematic Binder." (ICLR 2023)
> >
> > [3] Jiang et al., "Object-Centric Slot Diffusion." (arXiv 2023)

---

> > > ### Author Response · Authors · 2023-11-17
> > > **Response to Reviewer jXVc (3/3)**
> > >
> > > > This is also a manual effort, and arguably harder than a post-hoc approach?
> > >
> > > While it might appear that our methods involve manual effort, such as designing the data augmentation, we contend that this effort is negligible when compared to the post-hoc procedures required in previous methodologies.
> > >
> > > In the previous methods like IODINE, empirical studies during inference are crucial to establish relationships between specific values in a latent vector for an object and various properties like position, size, and color.
> > > To find the relations, one should traverse all values in a latent vector and also decode the vector into an image.
> > > Discovering these relationships involves traversing all values in a latent vector and decoding it into an image.
> > > Furthermore, ensuring consistent functionality of the relation function across different latent vectors or objects poses uncertainty, making the post-hoc process in previous methods cumbersome.
> > >
> > > In contrast, designing data augmentation demands minimal effort compared to the aforementioned post-hoc procedures.
> > > The process primarily involves incorporating random sampling procedures—such as translating, scaling, and tinting—during training.
> > > During inference, devoid of empirical studies on each slot-property relation, our method allows direct control over slots and their corresponding decoded images using instruction vectors, just as during training.
> > > In this context, we assert that our method achieves a more straightforward and interpretable controllability than previous methodologies, with minimal additional manual efforts.
> > >
> > > > Definition of "Durability Test”
> > >
> > > Firstly, we are grateful to the reviewer for recommending a more concrete expression of our terms.
> > > Incorporating equations, as suggested, can enhance the understandability of our experimental concept.
> > > However, as indicated by the reviewer, we maintain that our current definition of the task is also clear.
> > > And in fact, to fully describe the durability test, for instance the multi-step test, we need to articulate the process of forward and backward execution N times such as
> > >
> > > $(g_{1}g_{2}...g_{k})^{-1}(g_{1}g_{2}...g_{k})...
> > > (g_{1}g_{2}...g_{k})^{-1}(g_{1}g_{2}...g_{k})(x) = x$.
> > >
> > > Compared to this equation, we think that the current textual description may convey the information in a more concise manner.
> > > Still, we value the reviewer’s comment, encouraging us to explore alternative ways to articulate the definition of the durability test.
> > >
> > > Secondly, we have conducted an experiment to measure the equivariance in the both relationships $g_{1}(image) = h_{1}(slots)$ and $(g_{1}^{-1}g_{1})(image) = (h_{1}^{-1}h_{1})(slots)$, with the results presented in Table A in the appendix.
> > > In line with the reviewer's insightful suggestion, we acknowledge the value of conducting a durability test for the whole slots from an image while measuring the degree of the equivariance.
> > > We are working on implementing the suggested evaluation, however, constructing the benchmark and evaluating our models is taking longer than anticipated due to the limited availability of benchmark datasets at present.
> > > Therefore, it may not be feasible to incorporate these experiments in the upcoming revision during our rebuttal period.
> > > However, we are committed to including them in the subsequent future version of our paper.
> > >
> > > Again, we appreciate the insightful question and valuable suggestion by the reviewer.

---

> > > > ### Comment · Reviewer_jXVc · 2023-11-22
> > > >
> > > > I appreciate the authors detailed response, both the clear explanation of how they they see OCL as a fusion of symbolic and connectionist approaches in computer vision,  as well as the technical details and additional experiment provided.
> > > >
> > > > Overall I would say that the proposed method seems to be quite tailored to datasets with a clear notion of "objects", and where images can be generated after manipulating properties of the objects in the scene. As other reviewers have noted, this restricts the type of datasets where the method can be used. Other works in OCL (like SlotAttention) can be and are applied on more general datasets -- so this is more about the particular technique than OCL in general.
> > > >
> > > > This technique feels a bit like [Neural Module Networks](https://openaccess.thecvf.com/content_cvpr_2016/html/Andreas_Neural_Module_Networks_CVPR_2016_paper.html), which learn specific neural operators that correspond to linguistic rules and can be composed. In this case, we are learning specific operators that correspond to transformations of the objects. The trend in NLP (and ML in general) seemed to be that given enough data it is better to let NNs learn the rules, rather than hard-code them.
> > > >
> > > > But good experiments in real-world settings could convince me otherwise in this context. For an idea such as this, which requires + enforces a lot of structure, I would expect the results to be quite good on datasets that reflect that structure. I'd like to see results on real-world datasets where those assumptions might not be true.
> > > >
> > > > I recognize that other papers in OCL use CLEVR (and variants), and many papers also use MNIST (and variants). As a wise advisor once said -- "if it doesn't work on MNIST then it doesn't work at all, but it might fail to generalize beyond MNIST." I feel CLEVR is similar to MNIST of objects.

---

> > > > > ### Author Response · Authors · 2023-11-23
> > > > > **Response to Reviewer jXVc**
> > > > >
> > > > > Thank you for the insightful and constructive response provided by the reviewer.
> > > > > We are pleased that our response addressed some of the reviewer’s concerns and questions and we have this chance to discuss with the reviewer.
> > > > >
> > > > > > Real-world datasets.
> > > > >
> > > > > Regarding the reviewer’s point about datasets, extending our research to real-world datasets requires two essential components: 1) a more powerful object discovery model and 2) an improved decoding model.
> > > > >
> > > > > 1. Achieving a more potent object discovery model involves various approaches, primarily using supervision signals.
> > > > > While directly training the current model on the target dataset might yield promising results, it deviates from the task-agnostic learning of visual sensory input we aimed for, as discussed in the previous response (1/3).
> > > > > Secondly, the reviewer's mention of foundational models like SAM for segmentation mask extraction and spatial binding presents a method used to achieve the state-of-the-art.
> > > > > 2. To enhance our decoding model's capabilities, we may consider incorporating techniques from Generative Adversarial Networks (GANs) or Diffusion.
> > > > > Notably, very recent research such as LSD [1] and SlotDiffusion [2], both known as spotlight papers at NeurIPS 2023, are delving into incorporating diffusion into slots.
> > > > > Differing from concurrent studies on Slot + Diffusion, which concentrate on refining decoding performance in OCL model, our research emphasizes rethinking individual slot decoding within OCL specifically concerning the image-to-object loss separation.
> > > > > Combining our proposed aspects with Slot + Diffusion research could yield valuable results in the future.
> > > > >
> > > > > We recognize that these approaches offer promising directions for expanding OCL models including ours to real-world datasets, however, we consider these avenues to be beyond the scope of our current paper.
> > > > >
> > > > > [1] Jiang et al., “Object-centric slot diffusion.” (arXiv 2023)
> > > > >
> > > > > [2] Wu et al., “SlotDiffusion: Object-Centric Generative Modeling with Diffusion Models.” (arXiv 2023)
> > > > >
> > > > > > "tailored to datasets with a clear notion of objects."
> > > > >
> > > > > We agree with the comment about our work being "tailored to datasets with a clear notion of objects.”
> > > > > However, we would say that the mentioned property does align with the philosophy and goal of our research domain known as "Object-Centric" Learning (OCL), rooted in compositional understanding.
> > > > > However, we believe our contribution can extend beyond this domain’s boundaries the since the potential future directions of our research domain are not limited to this aspect.
> > > > >
> > > > > ---
> > > > >
> > > > > We believe our research can evolve and be more generally applicable, aligning and collaborating with other state-of-the-art techniques.
> > > > > While Slot Attention introduced a model architecture capable of generalization, we believe our research can shed light the possibility of object-level controllability in terms of latent vectors (slots) with interpretability and interactivity.
> > > > > We would like to highlight that our research is the first paper showing this aspect and signifies its value.
> > > > >
> > > > > Yes, it might fail to generalize beyond MNIST (CLEVR).
> > > > > However, “you only fail when you stop trying,” as one of the greatest scientists once said.
> > > > > Once again, we express our gratitude for the reviewer’s time and valuable comments.

---

### Official Review · Reviewer_dLyp · 2023-10-30

**Soundness:** 3 good
**Presentation:** 3 good
**Contribution:** 3 good
**Rating:** 8
**Confidence:** 4

**Summary:**

This paper introduces SlotAug, an object-centric learning method that allows for interpretable manipulation of the slots. The model is trained with image-level data augmentation and supports scaling, translating, or color shifting individual objects in the scene. The authors introduce the concept of sustainability, which refers to the ability to preserve the nature of the slots, allowing for multiple iterations of slot manipulations. To achieve sustainability, the authors incorporate two submethods, Auxiliary Identity Manipulation (AIM) and Slot Consistency Loss (SCLoss). In experiments on Tetrominoes, CLEVR6, CLEVRTex6, and PTR, the authors demonstrate the ability to manipulate slots and the effectiveness of their model in achieving sustainability of the slots.

**Strengths:**

This paper introduces a novel approach to the important problem of interpretable and controllable object representations. The idea of leveraging image-level augmentations to enable object-level controllability by taking advantage of the independence of the slots has not been done before, as far as I know. I found the paper generally well-written and easy to understand, although I do list some questions and suggestions regarding clarity below. The experiments clearly demonstrate the ability of their model to manipulate the slots and the benefits of AIM and SCLoss for improving sustainability. The results from section 4.3 are also encouraging in showing that this method potentially helps improve the representation quality of the slots themselves.

**Weaknesses:**

- In section 4.1.1, the authors claim that one of the reasons their method works is because of the spatial broadcast decoder independently decoding for each slot. This is supported in the appendix by an experiment on SLATE which uses a decoder where the slots are not completely independent. This seems potentially limiting as several recent works in scaling object-centric learning (OCL) to realistic scenes [2, 3, 4] rely on decoders where each slot may not be decoded independently. This may limit the applicability of this method to more realistic scenes that are supported by those OCL methods.
- The second claim in section 4.1.1, that the use of ARK is important, does not seem to be supported by any experiments. How well does this method work with vanilla slot attention? Is ARK required for this method to work?
- I could not find which datasets are used for Table 1 and section 4.3 (Table 3 and Figure 6). The fact that segmentation quality is maintained and representation quality potentially improved is an important result. I would be curious about these results broken down by datasets.
- In the appendix, the authors mention that the scaling augmentation takes into account the predicted attention maps between the encodings and the slots to handle the translation of objects during scaling. I am a bit confused about this. Does this mean that the augmentation changes as the model gets trained better? Or does this use some other pre-trained Slot Attention encoder?
- In terms of the presentation, I was initially unsure of the significance of sustainability until I saw the experimental results in section 4.2. For clarity, I would suggest showing a motivating example earlier in the text to explain the necessity of the AIM and SCLoss components.

**Questions:**

- Are the qualitative examples cherry-picked? Are there common failure scenarios the reader should be aware of?
- I am confused about the use of the SRT decoder in some of the experiments since that method does not have any notion of slots. Was this supposed to be the OSRT decoder [1]?


[1] Object Scene Representation Transformer. https://arxiv.org/abs/2206.06922

[2] Simple Unsupervised Object-Centric Learning for Complex and Naturalistic Videos. https://arxiv.org/abs/2205.14065

[3] Object-Centric Slot Diffusion. https://arxiv.org/abs/2303.10834

[4] SlotDiffusion: Object-Centric Generative Modeling with Diffusion Models. https://arxiv.org/abs/2305.11281

---

> ### Author Response · Authors · 2023-11-14
> **Response to Reviewer dLyp (1/2)**
>
> Appreciate the reviewer's thoughtful review and interest in our paper.
> We will respond to each of the comments and we hope to have more discussion about our work and the future direction of our research domain.
> Some responses still have pending points, such as ongoing experiments and figure creation, and we will do our best to resolve these aspects during this rebuttal period.
>
> > Constraints on the type of decoder may limit the applicability of this method.
>
> As the reviewer correctly noted, recent research has been integrating large decoders, allowing for interactions between slots to improve decoding quality. (In this response, we will refer to this approach as a "mixture decoder.")
>
> Taking a slightly different viewpoint from the aforementioned research, our study reevaluated the concept of independence decoding, as demonstrated by the spatial broadcast decoder proposed in the original Slot Attention. We believe that independent decoding holds value comparable to the excellent decoding quality achieved by mixture decoders. It's crucial to note that in this research, we empirically and theoretically demonstrated that independent decoding (not only the spatial broadcast decoder but also the transformer-based SRT decoder) divides image-level loss into object-level loss. However, this does not imply that mixture decoding cannot achieve the same; it's just that success is not guaranteed, as illustrated in the SLATE case in the appendix.
>
> Future research could explore how object-centric learning varies based on the decoder's configuration. Additionally, structuring a decoder with both mixture decoding and slot-wise decoding as two branches holds promise for achieving high-level decoding quality while learning controllability. We look forward to investigating these aspects in upcoming studies.
>
>
> > Is ARK required for this method to work?
>
> In fact, ARK is not necessary for our method. However, as shown in the SLASH paper, the original Slot Attention (SA) suffers from a bleeding issue, wherein the attention or segmentation masks for objects leak into the background. As we mentioned in the proof in the appendix, the performance of object discovery plays a significant role in object-level controllability. If our proposed training scenario arises where bleeding issues do not occur in the original SA, it can be achieved without the need for ARK.
>
> The use of ARK is not intended to enhance object discovery performance in a single training session; rather, it is employed to ensure consistent results across multiple experiments. Thanks to the reviewer’s feedback, we acknowledge the necessity of providing more detailed descriptions of these aspects, and also experimental results. We are conducting the experiments with the original Slot attention and will add the results with the descriptions in the revision during the rebuttal period.
>
> > The dataset used for Table 1.
>
> Thank the reviewer for finding the lack of description regarding the datasets used in our study. The results are from CLEVR6.
>
> As you have already figured out, the focus of our study is not to enhance the performance of the object discovery, and the experiment results support our claim that our method does not negatively impact the performance of the existing OCL model.
>
> Nevertheless, we also acknowledge the importance of including results from other datasets, such as CLEVRTEX and PTR, which would show the robustness and applicability of our method. We plan to carry out experiments on these datasets and intend to include these additional results in the appendix if the results are ready during the ongoing discussion. Even though we cannot include the experimental results in the revision, we will incorporate them in the following version of our paper.
>
> > Details about the scaling augmentation.
>
> We appreciate the reviewer for posing this question since this question, we believe, signifies the reviewer's dedication to comprehending our work in detail, and we value the effort invested in understanding our paper thoroughly.
>
> As the model undergoes training, we obtain more precise calibration results, corresponding to the more accurate predicted attention maps. However, the thing is that we do not use either the augmentation curriculum as the model gets trained better or the pre-trained encoder. Initially, we also considered equipping the model with object discovery and translation abilities before employing scaling augmentation during training. However, we found that a straightforward training curriculum, incorporating all types of augmentation (scaling, translating, and tinting) from the first epoch, proved effective without compromising performance in object discovery and manipulation. To maintain simplicity and clarity in presenting our work, we opted not to include any curriculum learning components.

---

> > ### Author Response · Authors · 2023-11-14
> > **Response to Reviewer dLyp (2/2)**
> >
> > > An earlier motivating example for sustainability.
> >
> > We fully agree with the reviewer's suggestion and are considering the incorporation of motivation for sustainability into Figure 1 in the main paper. We appreciate the reviewer's feedback and will update the figure in the upcoming revision to reflect this addition.
> >
> > > Failure cases.
> >
> > In addition to the failure examples provided in the extreme durability test section of the appendix in the submitted paper, we agree with the reviewer’s feedback on the general failure cases. We will append more failure examples in an upcoming revision and notify the reviewer of the update. We hope to give a deeper understanding of and an insight into our method.
> >
> > > The SRT decoder or the OSRT decoder?
> >
> > Thanks again for the reviewer's delicate question. We use SRT decoder rather than OSRT decoder. The reason we chose the SRT decoder is that it is simple and allows for independent decoding for each slot, while the OSRT decoder, due to its incorporation of "slot mixing," does not align with our requirement for independent decoding.
> >
> > The original SRT decoder was designed to operate at the image level, and the OSRT paper introduced a modification to decode slots simultaneously as shown in the Figure 2 in the OSRT paper. However, we slightly modified it to decode each slot independently following the spatial broadcast decoder.
> >
> > This selection was made to demonstrate that our proposed method is not limited to CNN-based spatial broadcast decoders used in Slot Attention but can robustly operate within transformer-based decoders as well, given the appropriate conditions for independence.
> >
> > We will add more details in the upcoming revision.

---

> > > ### Comment · Reviewer_dLyp · 2023-11-22
> > > **Response to Rebuttal**
> > >
> > > Thank you for taking the time to answer my questions. I think this is a good contribution to the object-centric learning community and have decided to increase my score to 8.

---

### Official Review · Reviewer_byFg · 2023-11-01

**Soundness:** 3 good
**Presentation:** 3 good
**Contribution:** 3 good
**Rating:** 6
**Confidence:** 4

**Summary:**

The paper studies slot-based unsupervised image models, e.g., Slot-Attention, and proposes a way to introduce controllability in the slots. This is done via enforcing a form of equivariance of the image -> slot transformation to image augmentations, except that the augmentations applied in the slot space are a learnable mapping from the augmentation instuctions. The results demonstrate that the model successfully manages to control and manipulate slots given the instructions, and gracefully handles the inverse instructions to "undo" the given manipulations.

**Strengths:**

- The paper proposes a very original way to manipulate learnable objet slots in slot attention.
- The main advantage of the method is its simplicity: the augmentations are introduced on the image level, removing the need to implement per-slot manipulation strategies at training, yet the manipulations can be applied to individual slots at inference time leaving the other slots intact.
- The qualitative results are very impressive, even thought the datasets are quite simple. The findings of the paper are encouraging for the future research on slot controlability.

**Weaknesses:**

The main weakness of the method is the fact requires a pair of (image augmentation, augmentation instruction) to work, rather than only one of them. Iit is easy to generate both the augmentation and its instruction with simple image transformation in a controlled environment, but this is much harder to do in a realistic setup. Some image transformation may not have a clear apriori-known instruction, or vice-versa, some may only have the instruction for the augmentation (e.g., specified as text) without the knowledge of the augmentation.
Eschewing this requirement would largely benefit the method and make it applicable in a more realistic setup.

**Questions:**

No questions

**Details Of Ethics Concerns:**

No ethic concerns

---

> ### Author Response · Authors · 2023-11-13
> **Response to Reviewer byFg**
>
> Thank the reviewer for both the positive and encouraging comments, and the insightful feedback on the weakness of our work, as well as the potential future direction of this area. We agree with the reviewer's observation regarding the requirement of pairs of (image augmentation, augmentation instruction).
>
> We recognize that there can be diverse, almost infinite, scenarios for controlling objects and object representations in distinctive domains. Our research, being in its early stages, considered relatively simple instructions and scenarios with clear relationships between instructions and object properties. As suggested by the reviewer, there is a need to address this limitation by advancing our research into more realistic settings. Therefore, future studies should focus on validating controllability in situations that are more practical and usable.
>
> Hopefully, recent research, such as [1, 2], demonstrates the potential of incorporating language data with slot representation. Furthermore, dataset papers like [3] offer promising directions in this field. In line with this, we are preparing subsequent research that utilizes caption datasets and visual-language models. We are also curious to explore the feasibility of interpretable controllability over object representation using single images and their corresponding descriptions, even without the need for pairs of (reference image, augmented/edited image) data.
>
> We appreciate the reviewer's recognition of the potential and contribution of our research despite the mentioned limitation of our study.
>
> [1] Kim et al., Improving Cross-Modal Retrieval with Set of Diverse Embeddings. (CVPR 2023)
>
> [2] Kim et al., Shatter and Gather: Learning Referring Image Segmentation with Text Supervision. (ICCV 2023)
>
> [3] Wang et al., The All-Seeing Project: Towards Panoptic Visual Recognition and Understanding of the Open World. (arXiv 2023)

---

### Author Response · Authors · 2023-11-13
**General Response**

We deeply appreciate the reviewers' thoughtful and constructive feedback. The positive comments, such as describing our approach as "**a very original way**" (byFg) and "**a novel approach**" (dLyp), acknowledging the "**impressive qualitative results**" (byFg), and recognizing the importance of our motivation, are truly motivating for us. We are encouraged by the reviewers’ remarks, including "**has not been done before**" (dLyp), "**encouraging for future research**" (dLyp), "**the main advantage is its simplicity**" (byFg), and the kind words about the clarity of our writing, "**well-written and easy to understand**" (dLyp),  and the "**well-organized experiments**" (jXVc).

We thank the reviewers for recognizing the motivation behind our paper, even though we acknowledge that our contribution represents just a small step in the longer journey towards achieving interpretable controllability in the artificial neural networks.

We're also grateful for the keen feedback on our areas of improvement and the insightful questions. We think that these questions from each reviewer cover various aspects, and we will respond to each one individually in a timely manner. The reviewers’ feedback is invaluable to us, and we look forward to addressing each comment and question to enhance the quality of our work.

---

### Author Response · Authors · 2023-11-19
**List of Revisions**

1. Added a reference sentence in relation to Figures 4 and 5, aiming to reinforce the earlier context of the sustainability concept in Section 1 of the main paper.
   - We deliberated on incorporating the suggested motivation (by Reviewer dLyp) into a new version of Figure 1. However, after careful consideration, we determined that adding additional content to the current Figure 1 might clutter the figure. Therefore, we opted to add a brief sentence to guide readers to Figures 4 and 5, which serve as the motivating visuals for understanding the sustainability concept. Any feedback on how we can efficiently provide the earlier motivation for the sustainability concept would be greatly appreciated.

2. Revised the expressions and added citations for the related works for the “spatial binding” in Section 3.

3. Included the indication of the used dataset in the caption of Table 1.

4. Added the details about our SRT decoder in Section C in the appendix.

5. Extended the analysis of the ARK in Section D and experimental results in Figure C, both in the appendix.

6. Added failure cases in Figure D in the appendix.

---

### Meta-Review · Area_Chair_gxVy · 2023-12-05

**Metareview:**

This paper proposes a method for learning and controlling object-centric scene representations in the context of generative image modeling. In particular, it builds on recent methods for unsupervised scene decomposition using latent slot-based representations and develops a scheme for enabling control of certain attributes of these representations by generating paired dataset examples of ground-truth scene edits.

The paper addresses a problem of relevance to the ICLR community and most of the reviewers agree that the paper is overall well-written and that it proposes an interesting, novel solution for this problem in the context of unsupervised learning.

There was generally some disagreement between the reviewers on how to judge the paper’s suitability for ICLR, which I would like to clear up in the following: the reviewers recommending acceptance highlight the novelty of the method and that it contributes successfully to an existing literature on unsupervised scene representation learning on synthetic multi-object images.

The reviewer recommending rejection argues that the paper is of low relevance to the overall vision or generative modeling community as it makes strong assumptions that likely only hold for synthetic data (i.e. it does not demonstrate any real-world results). The reviewer further raises concerns regarding the terminology used in the paper (e.g. introduction of terms like “sustainability” for a concept related to equivariance).

My recommendation for this paper is that it needs to convincingly demonstrate some path to transferring the presented editing capability to some form of real-world data (where precise scene edits are not available in the training set) to be of relevance to the broader community. This does not necessarily require training a model from scratch on real-world data, but demonstrating some form of transfer would be needed to rule out that the method has no utility beyond a very constrained synthetic data setup (which is a real concern). The experiments on synthetic datasets of higher complexity (CLEVRTex and PTR) certainly help strengthen the paper, but they do not resolve the primary underlying concern.

Overall, while I agree with all the positive points raised about the paper, I think this paper still needs more work before it can be accepted at a conference like ICLR.

**Justification For Why Not Higher Score:**

Unclear relevance to the broader community beyond a specific synthetic multi-object image setup. Recent work on unsupervised scene decomposition has moved beyond significantly the complexity of the datasets covered in this work and is applicable to real-world data.

**Justification For Why Not Lower Score:**

N/A

---

### Decision · Program_Chairs · 2024-01-16

Reject